# EMU: GENERATIVE PRETRAINING IN MULTIMODALITY

**Quan Sun**[1*]  **Qiying Yu**[2,1*]  **Yufeng Cui**[1*]  **Fan Zhang**[1*]  **Xiaosong Zhang**[1*]
**Yueze Wang**[1]  **Hongcheng Gao**[1]  **Jingjing Liu**[2]  **Tiejun Huang**[1,3]  **Xinlong Wang**[1†]

[1] Beijing Academy of Artificial Intelligence  [2] Tsinghua University  [3] Peking University

Code & Demo: `https://github.com/baaivision/Emu`

## ABSTRACT

We present **Emu**, a multimodal foundation model that seamlessly generates images and text in multimodal context. This omnivore model can take in any single-modality or multimodal data input indiscriminately (*e.g.*, interleaved image, text and video) through a one-model-for-all autoregressive training process. First, visual signals are encoded into embeddings, and together with text tokens form an interleaved input sequence. **Emu** is end-to-end trained with a unified objective of classifying the next text token or regressing the next visual embedding in the multimodal sequence. This versatile multimodality empowers the leverage of diverse pretraining data sources at scale, such as videos with interleaved frames and text, webpages with interleaved images and text, as well as web-scale image-text pairs and video-text pairs. **Emu** can serve as a generalist multimodal interface for both image-to-text and text-to-image tasks, supporting in-context image and text generation. Across a broad range of zero-shot/few-shot tasks including image captioning, visual question answering, video question answering and text-to-image generation, **Emu** demonstrates superb performance compared to state-of-the-art large multimodal models. Extended capabilities such as multimodal assistants via instruction tuning are also demonstrated with impressive performance.

## 1 INTRODUCTION

With text corpus at massive scale, Large Language Models (LLMs) (Brown et al., 2020; Chowdhery et al., 2022; Touvron et al., 2023) with straightforward training objectives such as next-word-prediction learn to understand, reason, and generate text with unprecedented accuracy and fluency, paving the way for diverse real-life applications (Schulman et al., 2022) unthinkable a decade ago. Recent studies (Alayrac et al., 2022; Driess et al., 2023; Hao et al., 2022) further investigate Large Multimodal Models (LMMs) beyond LLMs. Flamingo (Alayrac et al., 2022) connects a powerful language model with a pretrained vision encoder and inserts learnable layers to capture cross-modality dependencies, demonstrating strong abilities in multimodal zero-shot and in-context learning. Recent works (Li et al., 2023b; Dai et al., 2023; Huang et al., 2023; Liu et al., 2023b; Zhu et al., 2023a; Ye et al., 2023; Li et al., 2023a; Gong et al., 2023) adopt this framework and build LMM by docking a vision encoder with an LLM.

The prevailing training objective in such LMMs is predicting the next text token (Alayrac et al., 2022; Hao et al., 2022; Huang et al., 2023; Zhu et al., 2023a; Liu et al., 2023b; Li et al., 2023a), typically with a frozen vision encoder and no supervision for the vision part, which highly restricts model capacity. Besides, these LMMs are mostly trained on image-text pairs or documents, while overlooking video data as a potential scalable source of interleaved multimodal data. Documents interleaved with images (*e.g.*, textbooks, webpages) provide an intuitive representation of complex concepts, and have proved to be effective in empowering models with multimodal in-context learning ability (Alayrac et al., 2022; Zhu et al., 2023b). Videos, which usually contain interleaved image frames and subtitles (Figure 3), are an abundant source of multimodal data that naturally contains dense visual signals and encodes stronger cross-modal correlations with text than regular multimedia

---

*Equal contribution. † Correspondence to *wangxinlong@baai.ac.cn*.

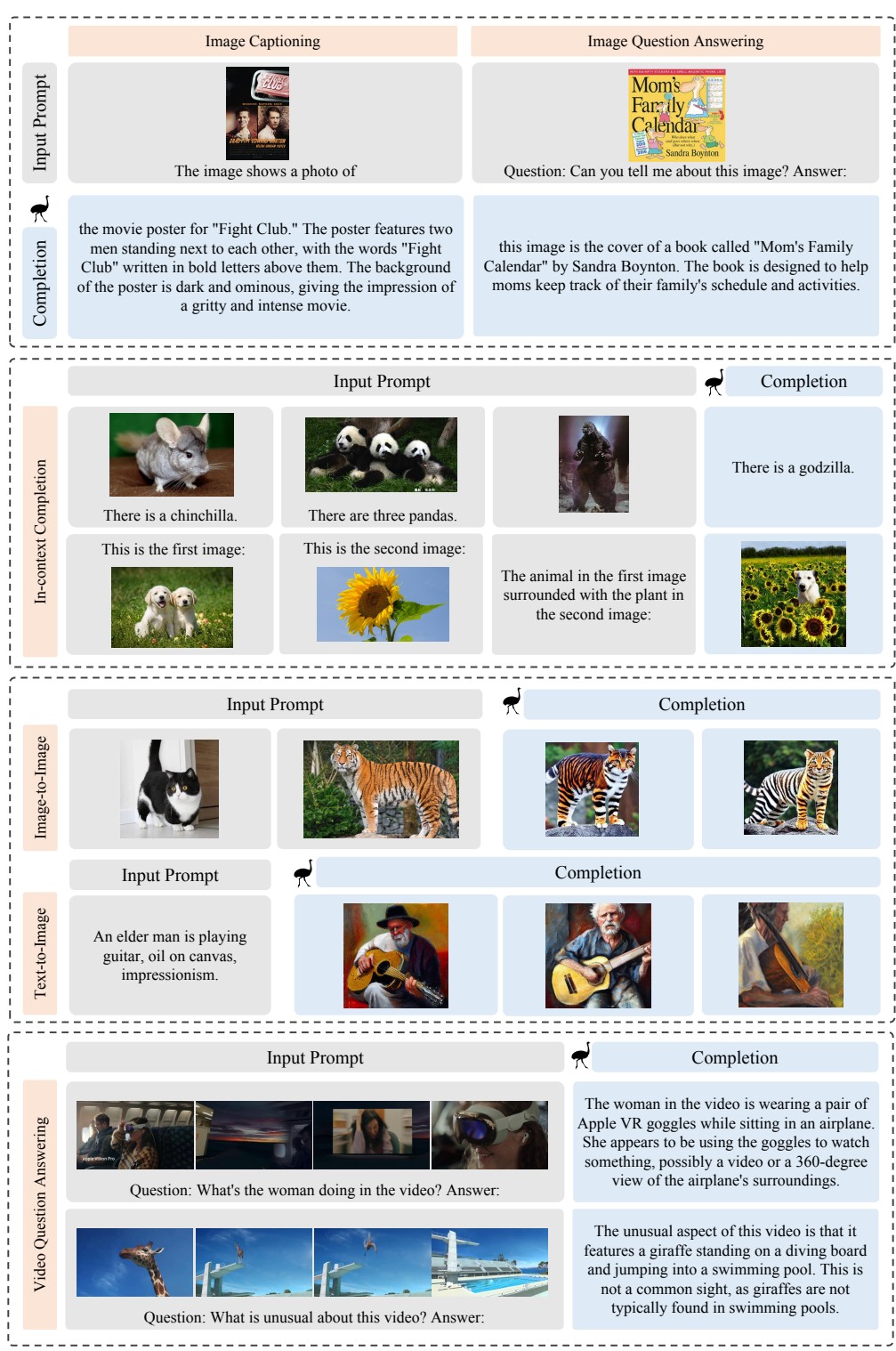

Figure 1: **Emu** as a generalist interface for diverse vision-language applications, such as image captioning, image/video question answering, in-context image-to-text and text-to-image generation, and image blending. More examples in Appendix E.

documents. Furthermore, public videos (especially user-generated clips) possess richer content diversity than Common Crawl[1], from which current training datasets mainly originate.

In this work, we introduce **Emu**, a large multimodal model that learns from both video and image data interleaved with text, under a unified objective of predicting the next visual or text token in an autoregressive fashion. To take advantage of rich web-scale data with an omnivore capacity, we formulate diverse sources of multimodal data (*e.g.*, videos with subtitles, webpages with images and text) into a unified format of interleaved image embeddings and text tokens (videos are converted into an interleaved sequence of randomly-selected frames and subtitles). Specifically, visual signals are first encoded into embeddings via a visual representation model EVA-CLIP (Sun et al., 2023), instead of being converted into discrete tokens. These visual embeddings together with text tokens constitute an interleaved multimodal input sequence, which will be fed into **Emu** for training.

We pretrain **Emu** on these multimodal data sequences under a simple unified objective: predicting the next element in a multimodal sequence. Different from existing LMMs that compute the predict-the-next loss on text tokens only, in training **Emu**, all input elements including both discrete text tokens and continuous image embeddings are accounted for loss computation. We adopt the cross-entropy classification loss for discrete text tokens, and the $\ell_2$ regression loss for continuous visual embeddings. As raw images typically lack the left-to-right causal dependency as in language, **Emu** does not perform image generative pretraining in the original pixel space. Instead, visual embeddings are transformed into a causal latent space via Causal Transformer, which accepts the image encodings generated by EVA-CLIP as input, and outputs $N$ tokens that capture the causal dependency of the given image (as illustrated in Figure 2).

Pretrained with the unified objective and diverse data forms, **Emu** can serve as a generalist interface for both image-to-text and text-to-image tasks by performing various types of completion in a multimodal sequence. As illustrated in Figure 1, **Emu** accepts multimodal prompts (*e.g.*, text, image, video, or their interleaved sequence) and generates multimodal response (for image generation, visual embeddings are decoded by a fine-tuned diffusion model). Further, **Emu** demonstrates impressive capabilities such as in-context text and image generation (the 2nd block of Figure 1), image blending (the 5th row of Figure 1 that combines a cat and a tiger into a real-looking tiger-cat), video understanding (the last block of Figure 1), and real-world knowledge grounding (Section 5.4).

We evaluate **Emu** on a broad range of zero-shot and few-shot tasks including image captioning, visual question answering, video question answering, and text-to-image generation. For qualitative demonstration, we also build an effective multimodal assistant via instruction tuning on multimodal conversational data. The instruction-tuned **Emu** assistant can effectively follow human instructions and interact with users via multimodal response.

## 2 EMU: PREDICT THE NEXT IN MULTIMODALITY

### 2.1 ARCHITECTURE

**Emu** is a large-scale multimodal model that performs completion in multimodality, *i.e.*, perceiving interleaved multimodal input and generating outputs varying in modalities. As illustrated in Figure 2, **Emu** consists of four parts: Visual Encoder, Causal Transformer, Multimodal Modeling, and Visual Decoder. We leverage pretrained EVA-CLIP (Sun et al., 2023), LLaMA (Touvron et al., 2023) and Stable Diffusion (Rombach et al., 2022) to initialize the Visual Encoder, the Multimodal Modeling LLM and the Visual Decoder, respectively.

Given any sequence of interleaved image, text and video, we first encode the image into dense visual features via EVA-CLIP, then transform the encodings into a fixed number of $N$ visual causal embeddings via Causal Transformer. Similarly, we encode a video of $T$ frames into $T \times N$ visual causal embeddings. Two special image tokens `[IMG]` and `[/IMG]` are prepended and appended for each image or frame, respectively, to represent the beginning and end of the encoded image/frame embeddings. The visual causal embeddings are combined with text tokens to form multimodal sequences that are fed into the Multimodal Modeling LLM for unified autoregressive modeling. We append `` and `` tokens to the start and the end of each sequence. In inference, we fine-tune the Visual Decoder to decode the visual embeddings into a realistic image.

---

[1]https://commoncrawl.org/

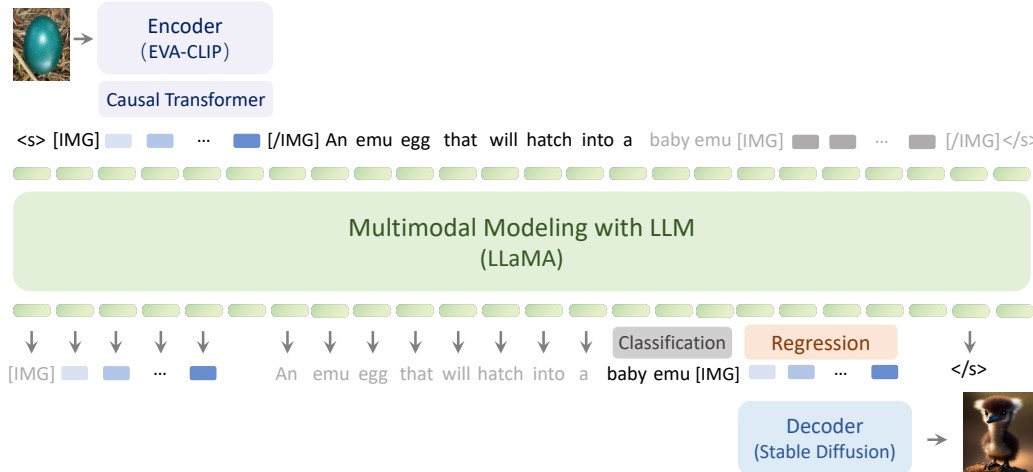

Figure 2: **Emu** unifies the modeling of different modalities in an auto-regressive manner. Visual signals are first encoded into embeddings, and together with text tokens form an interleaved sequence. The training objective is to either classify the next text token or regress the next visual embedding. In inference, regressed visual embeddings are decoded into a realistic image via a fine-tuned latent diffusion model.

**Causal Transformer.** Auto-regressively modeling images in raster order is counter-intuitive and has not demonstrated satisfactory performance, which may be attributed to the fact that images naturally possess 2D structures and are not perceived as sequential signals like text. To better capture the characteristics of images and achieve unified modeling of different modalities, we propose a Causal Transformer module to transform 2D spatial visual signals to 1D causal sequences in a latent space $Z$. Specifically, given an image $I$ with its encodings $g(I)$ from EVA-CLIP as condition, Causal Transformer accepts randomly initialized embeddings $\{e_1, e_2, \ldots, e_N\}$ as input, and outputs $N$ embeddings $\{z_1, z_2, \ldots, z_N\}$ that capture the causal dependency of the given image. The architecture of Causal Transformer is similar to the decoder of Transformer (Vaswani et al., 2017), with each block consisting of a causal self-attention layer, a cross-attention layer, and a feed-forward layer. The cross-attention layer aggregates visual information from the image embeddings extracted from EVA-CLIP, where the visual embeddings are treated as keys and values, and the outputs from the previous causal attention layer serve as queries.

**Visual Decoder.** We use a latent diffusion model to decode visual embeddings into images, and adopt the weights of Stable Diffusion (Rombach et al., 2022) for initialization. Specifically, we feed $N$ visual embeddings generated by **Emu** into the diffusion model as conditions for image decoding. We replace the linear projections of the cross-attention modules in Stable Diffusion with new linear layers that accommodate the dimension of **Emu** and Stable Diffusion.

## 2.2 TRAINING OBJECTIVE

Given an unlabeled web-scale corpora $\mathcal{D}$ consisting of interleaved multimodal sequences $x = (x_1, x_2, \ldots, x_n)$, where $x$ can be vision-language sequences of various forms, such as image-text pairs, image-text interleaved documents, or videos with subtitles. $x_i$ can be a signal unit (text or image token) from any arbitrary modality. We first convert all continuous 2D signals (images and video frames) into 1D causal latent embedding sequences using Causal Transformer, then insert them back into the corresponding places in the sequence $x$. The resulting sequence is represented as $u = (u_1, u_2, \ldots, u_m)$, where $u_i$ can be either a discrete text token, or a visual embedding that captures causal dependency with neighboring visual embeddings.

We approximate the likelihood of the web-scale corpora $p(x)$ with $p(u)$, and maximize the likelihood in a unified auto-regressive manner as follows:

$$\max_\theta \sum_{u \in \mathcal{D}} \sum_{i=1}^{|u|} \log P(u_i | u_1, \ldots, u_{i-1}; \theta) \approx p(x) \tag{1}$$

Two types of losses are adopted to optimize this objective. For discrete text tokens, cross-entropy loss is used to supervise classification in the predefined vocabulary with a language modeling head. For continuous visual embeddings, $\ell_2$ regression loss is adopted with a separate regression head.

## 2.3 GENERALIST INTERFACE

The unified auto-regressive modeling of different modalities endows **Emu** with a powerful ability to serve as a multimodal generalist that can perform any types of completion in a multimodal sequence, *i.e.*, accepting multimodal sequence as input, and outputting signals across vision and language modalities. For example, given two examples as the prompt, **Emu** automatically infers and completes the corresponding task given a new input, as shown in the second block of Figure 1.

Specifically, given a multimodal context, if the expected output format is text, **Emu** will use the language modeling head to generate discrete text tokens. If the desired output is image, we will append a `[IMG]` token at the end of the input sequence, then **Emu** will autoregressively generate $N$ visual embeddings that will then be sent to the visual decoder for decoding into a real-world image.

## 3 EMU TRAINING

We pretrain **Emu** with web-scale data across modalities in various forms, including image-text pairs (LAION-2B (Schuhmann et al., 2022), LAION-COCO (lai, b)), interleaved images-text data (MMC4 (Zhu et al., 2023b)), video-text pairs (WebVid-10M (Bain et al., 2021)), and our collected interleaved video-text data (YT-Storyboard-1B). All these data are formulated as multimodal sequences, from which **Emu** learns under the objective of predict-the-next-element in a unified auto-regressive manner. After pretraining, we finetune a decoder to transform visual embeddings into realistic images.

### 3.1 DATA

**Image-text Pairs.** We use the image-text pairs from LAION-2B (Schuhmann et al., 2022) and LAION-COCO (lai, b) for pretraining. LAION-2B provides images paired with noisy alt-texts from the web, and LAION-COCO is its 600M subset that is captioned by BLIP (Li et al., 2022).

**Video-text Pairs.** WebVid-10M (Bain et al., 2021) is an extensive dataset consisting of a large collection of short videos with textual descriptions. These videos are sourced from materials websites with diverse contents and a strong correlation between text and video. We use heuristic rules to remove irrelevant metadata (*e.g.*resolution of the original video, camera parameters).

**Interleaved Image and Text.** Large-scale image-text interleaved data plays a crucial role in unlocking the in-context learning ability of multimodal models. We leverage the Multimodal-C4 (MMC4) dataset (Zhu et al., 2023b), an expanded version of the text-only C4 (Raffel et al., 2020). MMC4 comprises a collection of approximately 75 million image-text-interleaved documents, with 400 million images and 38 billion tokens in total. From each document, we sample a random subsequence of L = 1024 take up to the first N = 5 images. Additionally, we randomly sample N = 5 images along with their corresponding sentences to construct a subsequence of L = 512.,

**Interleaved Video and Text.** Videos with subtitles also present a promising and scalable source of interleaved multimodal data. We introduce the YT-Storyboard-1B dataset which collects 18 million videos and their corresponding subtitles from YouTube[2] using the video-ids provided by the YT-Temporal-1B dataset (Zellers et al., 2022). Instead of raw videos, we collect storyboard images (about 1.8 billion images in total), a set of thumbnails provided by the YouTube website for quick video viewing. The combination of storyboard thumbnails and subtitles creates a natural interleaved sequence of video and text ordered by timestamps, as in Figure 3. More details are in Appendix A.1.1.

### 3.2 PRETRAINING

We initialize **Emu**'s Visual Encoder with the 1B version of EVA-01-CLIP (Sun et al., 2023), and Multimodal Modeling LLM with the 13B version of LLaMA (Touvron et al., 2023). LLaMA is a decoder-only Transformer (Vaswani et al., 2017) and EVA-01-CLIP is a 40-layer ViT (Dosovitskiy

---

[2]`https://www.youtube.com`

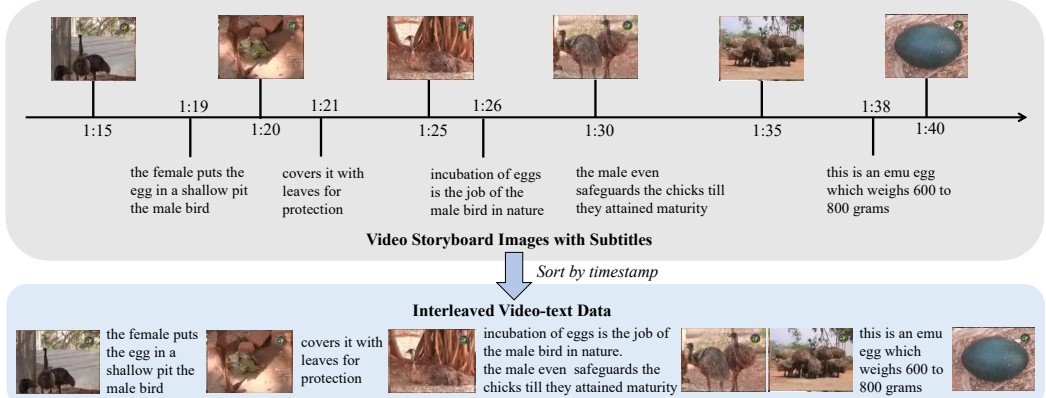

Figure 3: Interleaved video-text data. The combination of storyboard thumbnails and subtitles captions creates a natural interleaved sequence of video and text that is ordered by the timestamps.

et al., 2021). The Causal Transformer comprises 12 blocks, each of which consists of a causal self-attention layer, a cross-attention layer, and a feed-forward layer. Random initialization is used for Causal Transformer. The total number of parameters of **Emu** is 14B and is trained end-to-end.

We use a batch size of 128 for image-text pair data, 64 for interleaved image-text data, 16 for video-text pair and interleaved video-text data. Detailed pertaining hyperparameters are in Appendix A.1.1. For each video, we randomly sample 8 frames for pretraining, and all images/frames are resized into 224×224 resolution. For image-text pair and interleaved data, we randomly put each image before or after its corresponding sentence. We train the model on 128 NVIDIA 80G-A100 GPUs for 10k steps with around 82M samples (150B tokens in total), and the pretraining takes approximately 2 days.

## 3.3 VISUAL DECODING

After pretraining, we tune the visual decoder with both LAION-COCO (lai, b) and LAION-Aesthetics (lai, a) (a high-aesthetics quality subset of LAION-5B (Schuhmann et al., 2022)) image-text pair datasets under text-to-image task. Specifically, We initialize the diffusion model with Stable Diffusion v1.5. We freeze the Visual Encoder, Multimodal Modeling LLM in **Emu**, and the VAE in diffusion model during training, with only the parameters of U-Net updated. For each training sample, we append the [IMG] token to the end of the input text and feed it into the Multimodal Modeling LLM, which will then generate $N$ visual embeddings in an auto-regressive manner. These visual causal embeddings are fed into Image Decoder as the condition for image generation training.

We follow the model setups of Stable Diffusion v1.5. We train the diffusion model with 32 A100-40G GPUs for 15k iterations. Detailed hyperparameters are in Appendix A.2. To further improve sample quality, we randomly drop image embeddings condition by $10\%$ of the time during training to enable classifier-free guidance (Ho & Salimans, 2022).

## 4 INSTRUCTION TUNING

Language instruction tuning has helped pretrained language models to align with user intentions (Ouyang et al., 2022; Wang et al., 2022c; Taori et al., 2023; Zheng et al., 2023) and generalize to unseen tasks (Wei et al., 2022; Chung et al., 2022). We apply multimodal instruction tuning on **Emu** to align it with human instructions through supervised finetuning on publicly available datasets, including language instructions from ShareGPT (Zheng et al., 2023) and Alpaca (Taori et al., 2023), image-text instructions from LLaVA (Liu et al., 2023b), and video instructions from VideoChat (Li et al., 2023c) and Video-ChatGPT (Maaz et al., 2023). Dataset details can be found in Appendix B.1.

In instruction tuning, we freeze all parameters of pretrained **Emu**, and fine-tune a low-rank adaption (LoRA) module (Hu et al., 2022). The main focus of instruction tuning is to align the model with natural language instructions, which are less relevant to vision features. Thus, we attach LoRA modules only to the self-attention layers of the Multimodal Modeling LLM, and add no adaptation to the Vision Encoder. Training details can be found in Appendix B.1.

Table 1: Zero-shot comparison, * indicates that the zero-shot prompt is built by using two examples from the task, where their corresponding images have been removed. **Emu-I** is the instruction-tuned **Emu** model. The best results are **bold** and the second best are underlined.

| Models | Image-Text Tasks | | | | | | | Video-Text Tasks | | |
|---|---|---|---|---|---|---|---|---|---|---|
| | COCO | NoCaps | Flickr30K | VQAv2 | OKVQA | VizWiz | VisDial | MSVDQA | MSRVTTQA | NExTQA |
| *Per-task Finetuning* | | | | | | | | | | |
| PALI-X-55B | 149.2 | 126.3 | - | 86.0 | 66.1 | 70.9 | - | - | 47.1 | 38.3 |
| MetaLM | 82.2 | 58.7 | 43.3 | 41.1 | 11.4 | - | - | - | - | - |
| Kosmos-1 | 84.7 | - | 67.1 | 51.0 | - | 29.2 | - | - | - | - |
| Flamingo-9B * | 79.4 | - | 61.5 | 51.8 | 44.7 | 28.8 | 48.0 | 30.2 | 13.7 | 23.0 |
| **Emu** | 112.4 | 96.5 | 72.0 | 52.0 | 38.2 | 34.2 | 47.4 | 18.8 | 8.3 | 19.6 |
| **Emu** * | - | - | - | 52.9 | 42.8 | 34.4 | 47.8 | 34.3 | 17.8 | **23.4** |
| **Emu-I** | 120.4 | 108.8 | 77.4 | 57.2 | 43.4 | 32.2 | 43.0 | 34.6 | 16.8 | 5.8 |
| **Emu-I** * | - | - | - | **62.0** | **49.2** | **38.3** | **51.1** | **37.0** | **21.2** | 19.9 |

All instruction-tuning data are packed with this template:

$$\texttt{<System Message>} \quad \texttt{[USER]:} \quad \texttt{<Instruction>} \quad \texttt{[ASSISTANT]:} \quad \texttt{<Answer>}, \quad (2)$$

where `[USER]` and `[ASSISTANT]` are special tokens initialized from the embeddings of words 'user' and 'assistant', respectively. `<System Message>` varies depending on the specific task (Appendix B.2). `<Instruction>` and `<Answer>` are actual slots for human instructions and assistant answers, and only `<Answer>` is accounted for loss computation.

## 5 EVALUATION

We evaluate **Emu** on a broad range of vision-language tasks including image captioning (MS-COCO (Chen et al., 2015)), image question answering (VQAv2 (Goyal et al., 2017), OKVQA (Marino et al., 2019), VizWiz (Gurari et al., 2018)), visual dialog (VisDial (Das et al., 2017)), video question answering (MSRVTTQA (Xu et al., 2017), MSVDQA (Xu et al., 2017), NextQA (Xiao et al., 2021)) and text2image generation(MS-COCO (Lin et al., 2014)). Details are described in Appendix C.1. We evaluate our pretrained and instruction-tuned models in zero-shot and few-shot settings.

### 5.1 ZERO-SHOT EVALUATION

In the zero-shot setting, the model is tested on tasks and datasets never encountered during training. Task-specific prompts are used to indicate different tasks to perform, without any additional tuning for model parameters.

**Multimodal Understanding.** Table 1 presents the zero-shot multimodal understanding performance of **Emu** and **Emu-I** (the instruction-tuned model). For zero-shot evaluation of **Emu**, we adopt the multimodal Chain-of-Thought prompting technique following Huang et al. (2023), which first asks the model to generate a caption for visual content before outputting the final result. Additionally, we evaluate using the same strategy following Flamingo (Alayrac et al., 2022), where two text-only examples from the task are used as prompts (results indicated by an *). For more detailed information regarding the evaluation, please refer to Appendix C.2.

On COCO captioning task, **Emu** achieves impressive zero-shot CIDEr score (Vedantam et al., 2015) of 112.4, which outperforms other LMMs by a large margin. In a wide range of image and video question answering tasks, **Emu** consistently surpasses LMMs like Kosmos-1 and Flamingo-9B. Notably, **Emu** achieves an accuracy of 34.4% on the complex VizWiz VQA dataset, versus Kosmos-1's 29.2% and Flamingo-9B's 28.8%. **Emu-I** is the instruction-tuned **Emu** model that achieves notable improvements. Remarkably, even with only 14B parameters, **Emu-I** can outperform much larger-scale Flamingo-80B model in several tasks such as VQAv2 (62.0% vs. 56.3%), VizWiz (38.3% vs. 31.6%), and MSVDQA (37.0% vs. 35.6%).

**Text2image Generation.** We evaluate the zero-shot image generation ability on the validation set of MS-COCO (Lin et al., 2014). Following (Ramesh et al., 2021), we randomly sample 30k prompts from the validation set and calculate the zero-shot FID (Heusel et al., 2017). The results are shown in Table 2. For the generation of both **Emu** and SDv1.5, we use PNDM (Liu et al., 2022) scheduler with

Table 3: Few-shot comparison. $k$ is the number of in-context examples, and we used the same example selection approach (*i.e.* RICES (Yang et al., 2022b) ) as Flamingo (Alayrac et al., 2022).

| Models | VQAv2 | | | VizWiz | | | MSVDQA | | | MSRVTTQA | | |
|---|---|---|---|---|---|---|---|---|---|---|---|---|
| | $k$=2 | $k$=4 | $k$=8 | $k$=2 | $k$=4 | $k$=8 | $k$=2 | $k$=4 | $k$=8 | $k$=2 | $k$=4 | $k$=8 |
| Kosmos-1 | 51.4 | 51.8 | 51.4 | 31.4 | 35.3 | 39.0 | - | - | - | - | - | - |
| Flamingo-9B | - | 56.3 | 58.0 | - | 34.9 | 39.4 | - | 36.2 | **40.8** | - | 18.2 | 23.9 |
| PALI-X | - | 56.9 | 57.1 | | | | | | | | | |
| **Emu** | **56.4** | **58.4** | **59.0** | **37.8** | **41.3** | **43.9** | **36.0** | **37.1** | 39.8 | **21.2** | **21.8** | **24.1** |

Table 4: Zero-shot evaluation regarding each core VL capability of MM-Vet (Yu et al., 2023b).

| Model | Rec | OCR | Know | Gen | Spat | Math | Total |
|---|---|---|---|---|---|---|---|
| LLaMA-Adapter v2-7B (Gao et al., 2023) | 16.8 | 7.8 | 2.5 | 3.0 | 16.6 | 4.4 | 13.6±0.2 |
| MiniGPT-4-14B (Zhu et al., 2023a) | 29.9 | 16.1 | 20.4 | 22.1 | 22.2 | 3.8 | 24.4±0.4 |
| InstructBLIP-14B (Dai et al., 2023) | 30.8 | 16.0 | 9.8 | 9.0 | 21.1 | 10.5 | 25.6±0.3 |
| DreamLLM-7B (Dong et al., 2023) | 41.8 | 26.4 | 33.4 | 33.0 | 31.0 | 11.5 | 35.9±0.1 |
| LLaVA-65B (Lu et al., 2023) | 39.2 | **28.2** | 26.2 | 28.3 | **33.0** | **15.0** | 35.5±0.3 |
| **Emu-I -14B** | **45.5** | 19.2 | **36.7** | **35.9** | 25.2 | 3.8 | **36.3±0.3** |

50 steps. We also adopt classifier-free guidance (Ho & Salimans, 2022) for better generation quality. The scaling factor is set to 5.0 and 3.0 for **Emu** and SDv1.5 respectively, as these settings yield the best performance for both models. **Emu** achieves better performance compared to a concurrent work GILL (Koh et al., 2023a), which also generates images with LLMs. However, our model is inferior to SDv1.5 in terms of FID. This is probably because the condition space (image embeddings) of our visual decoder deviates a lot from the condition space (text embeddings) of the diffusion model used as initialization, and our model is trained for a relatively short 15k steps.

## 5.2 FEW-SHOT EVALUATION

In few-shot evaluation, the model is prompted with task-specific prompts and several examples collected from the training data to evaluate its in-context learning ability. Evaluation details can be found in Appendix C.3. Table 3 presents the performance of the pretraining model **Emu** in image and video question answering tasks under the few-shot ($k = 2, 4, 8$) evaluation setting. We use the Retrieval In-Context Example Selection (Yang et al., 2022b) approach following Flamingo (Alayrac et al., 2022). **Emu** demonstrates superior performance to Flamingo-9B and Kosmos-1 under almost all scenarios. For example, **Emu** achieves a VQAv2 accuracy of 58.4% and VizWiz 41.3% under the 4-shot setting, surpassing Flamingo-9B by +2.1% and +6.4%, respectively.

Table 2: Zero-shot text-to-image results on MS-COCO validation set. 30k samples are randomly sampled for evaluation.

| Models | FID($\downarrow$) |
|---|---|
| *unimodal generation models* | |
| GLIDE (Nichol et al., 2021) | 12.24 |
| Make-A-Scene (Gafni et al., 2022) | 11.84 |
| DALL-E 2 (Ramesh et al., 2022) | 10.39 |
| SDv1.5 (Rombach et al., 2022) | 9.93 |
| Imagen (Saharia et al., 2022) | 7.27 |
| Parti (Yu et al., 2022b) | 7.23 |
| *multimodal generation models* | |
| GILL (Koh et al., 2023a) | 12.20 |
| **Emu** (ours) | **11.66** |

For video-text tasks, **Emu** demonstrates strong performance as well, such as 4-shot 21.8% v.s. Flamingo's 18.2% on the MSRVTTQA benchmark. Additionally, we can observe a positive correlation between the number of shots $k$ ($k = 0, 2, 4, 8$) and the performance of **Emu**. These results demonstrate **Emu**'s remarkable in-context learning ability.

## 5.3 IN-THE-WILD EVALUATION

Table 4 presents zero-shot evaluation results on the in-the-wild benchmark MM-Vet (Yu et al., 2023b). We report the mean and std of 5 evaluation runs following Yu et al. (2023b). For each core capability, the average score is reported. **Emu-I** exhibits state-of-the-art in-the-wild capability, and even outperforms LLaVA-65B (Lu et al., 2023) in Rec, Know, Gen abilities and the total score.

## 5.4 QUALITATIVE EVALUATION

Beyond quantitative benchmarks, we conduct adequate qualitative evaluation of **Emu**. **Emu** demonstrates impressive capabilities that cannot be evaluated on standard benchmarks, including real-world

knowledge grounding (upper right of Figure 4), interleaved multi-image understanding (left side of Figure 4), detailed video understanding (lower right of Figure 4), multimodal assistant (Figure 5), multi-turn dialogue (Figure 6), image blending (Figure 7), and (in-context) text-to-image generation. For in-context text-to-image generation, **Emu** can generate context-related images (in the first two rows of Figure 8, the generated images share the oil painting style in context, compared with the corresponding images generated without context in the first two rows of Figure 9), and follow context-related instructions, as shown in the 4th row of Figure 1. The multimodal in-context ability of **Emu** is responsible for this brand-new ability of image generation.

We also compare **Emu** with other state-of-the-art multimodal assistants in terms of the ability to perform typical image captioning tasks (Figure 10) and follow human instructions (Figure 11). In Figure 11, we test a slightly difficult instruction, and only **Emu** response properly to list 8 books written by Agatha Christie and then recommend one.

## 6    RELATED WORK

Multimodal pretraining (Radford et al., 2021; Jia et al., 2021; Sun et al., 2023; Chen et al., 2020; Kim et al., 2021; Wang et al., 2022d;a;b; Cho et al., 2021; Li et al., 2021; Yu et al., 2022a; Chen et al., 2023c; Lu et al., 2022) learns cross-modal interactions from large-scale multimodal data. Flamingo (Alayrac et al., 2022) bridges powerful yet private pretrained vision and large language models and first demonstrates remarkable multimodal zero-shot and few-shot behaviors. With the increasing impact (Schulman et al., 2022) and accessability (Touvron et al., 2023) of LLMs, recent work has also considered building multimodal models based on LLMs (Li et al., 2023b; Driess et al., 2023; Huang et al., 2023; Dai et al., 2023; Ye et al., 2023; Zeng et al., 2023; Koh et al., 2023b), such as BLIP-series (Li et al., 2023b; Dai et al., 2023) that connect frozen vision and language pretrained models with a Q-Former to bridge the modality gap. These LMMs commonly use predicting the next text token as the training objective and exert no supervision for vision data (Hao et al., 2022; Huang et al., 2023; Zhu et al., 2023a; Liu et al., 2023b; Ye et al., 2023). Instead, **Emu** unifies the modeling of vision and language with the objective of predicting the next visual or text token in an autoregressive manner, and further explores videos as a new source of interleaved image-text data. This unified modeling leads to a generalist interface for diverse multimodal tasks that output either image or text. Emerging recent studies (Zhu et al., 2023a; Liu et al., 2023b; Maaz et al., 2023; Li et al., 2023c; Liu et al., 2023a; Li et al., 2023a; Chen et al., 2023b;a) attempt to build powerful visual multimodal assistants based on LMMs through constructed conversation data. We also instruction-tune **Emu** using publicly available datasets and build a multimodal assistant that aligns well with human instructions on both images and videos.

## 7    LIMITATIONS AND FUTURE TOPICS

Emu shares the well-acknowledged constraints inherent in other LLMs and LMMs, including susceptibility to both visual and language hallucinations, slow auto-regressive inference speed, a cessation of knowledge updates after pretraining, and a potential for generating non-factual content. Besides, Emu predominantly focused on English-language data. As a result, the model's proficiency in languages other than English is currently delicate, and users should exercise caution when applying it in such contexts. Addressing challenges related to hallucination, enhancing inference speed, and expanding multilingual capabilities are crucial areas for future exploration and improvement.

## 8    CONCLUSION

In this work, we present **Emu**, a Large Multimodal Model trained with a unified autoregressive objective of predicting the next element, including both visual and textual tokens. Apart from commonly used image-text pairs and interleaved documents, we explore another scalable data source of image-text interleaved data, *i.e.*, video. **Emu** trained under such unified objective and diverse data can serve as a generalist interface that is capable of performing diverse multimodal tasks, such as image captioning, image/video question answering, and text-to-image generation, together with new abilities like in-context text and image generation, and image blending. We also build a multimodal assistant instruction-tuned on **Emu**, which exhibits excellent human-aligned abilities such as multi-turn dialogue. We hope that our work will inspire the community to continue exploring the potential of diverse multimodal data at web-scale and also the generative pretraining beyond vision and language.

## ETHICS STATEMENTS

Emu is currently in a preliminary stage and has been developed solely for research purposes. Its usage in specific applications is not recommended until comprehensive risk analyses have been conducted and corresponding mitigation strategies thoroughly explored. The ensuing discussion outlines potential risks and corresponding mitigation strategies of Emu, acknowledging the necessity for further research efforts to comprehensively assess associated risks.

### POTENTIAL RISKS

The ethical considerations associated with Emu primarily stem from two key aspects: 1) model initialization: the Multimodal Modeling module of Emu is initialized from an open-sourced large language model LLaMA (Touvron et al., 2023), the Visual Decoder module is initialized from Stable Diffusion (Rombach et al., 2022), and the Vision Encoder is initialized from EVA-CLIP (Sun et al., 2023). Consequently, Emu inherits the potential risks of generating harmful and biased information, including offensive language, propagation of social biases and stereotypes, and the generation of inappropriate content such as pornography and child abuse. 2) Pretraining data. The pretraining data of Emu are publicly available and they are sourced from the Internet, where bias and harmful information are prevalent. Besides, the datasets sourced from the Internet (such as Common Crawl) may include links to images with personal information, potentially compromising privacy and containing sensitive content like faces, medical images, or other personal data.

### MITIGATION STRATEGIES

It is crucial to reiterate that Emu is designed exclusively for preliminary academic research and should not be deployed in specific applications without rigorous risk analyses and mitigation strategy exploration. Deployment in production environments warrants a more thorough investigation into model behavior and potential biases.

Given the extensive size of pre-training datasets and the associated training costs, curating datasets and developing models for widespread use exceeds the scope of a single research paper. However, we are open to discussing mitigation strategies to help address ethical concerns.

Short-term approaches include: 1) relying on prompting to mitigate any biases and harmful outputs, 2) implementing rule-based filtering, human oversight, and evaluation to identify and block harmful information, 3) employing a discriminator model capable of classifying harmful information for enhanced blocking, 4) Emu itself can be finetuned to become a multimodal discriminator.

In the long term, strategies involve: 1) social or public policy interventions, such as regulatory frameworks and guidelines; 2) thoughtful product design, especially regarding user interface decisions; 3) advancements in AI Ethics of powerful large models, including the development of better benchmarks and improved mitigation strategies.

Additionally, to address privacy concerns, methods exist for obfuscating or generating personal human attributes like faces (Yang et al., 2022a; Maximov et al., 2020), ensuring anonymity without compromising the quality of learned representations. While this avenue is worth exploring, it is currently beyond the scope of this paper.

In conclusion, Emu is presently a model intended for preliminary research purposes only, and deployment should be deferred until the aforementioned issues are thoroughly considered and addressed. Caution must be exercised before transitioning to production environments.

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

# A EMU TRAINING

## A.1 PRETRAINING

### A.1.1 DATASET DETAILS

**Image-text Pairs.** The LAION-2B dataset is the english subset of Laion-5B (Schuhmann et al., 2022) and contains large-scale image-text pairs data. LAION-COCO (lai, b) is captioned 600M images from LAION-2B with an ensemble of BLIP (Li et al., 2022) and CLIP (Radford et al., 2021) models. Whereas the text in LAION-COCO (lai, b) exhibits enhanced fluency and relevance to the associated images, it has insufficient text diversity and a potential loss of high-level semantic information, including world knowledge contents presented in the original LAION-2B dataset. Thus, we employ both the LAION-2B and LAION-COCO (lai, b) datasets during **Emu** pretraining.

**Video-text Pairs.** Webvid-10M (Bain et al., 2021) dataset contains a diversity of content with strong correlation between text and video. However, we found that a certain amount of the data contained irrelevant metadata information (*e.g.*resolution of the original video, camera parameters). To prevent the model from being influenced by these irrelevant details, we use heuristic rules to remove these content. Firstly, we build a word list consisting of irrelevant information. This word list is then utilized as a filtering mechanism to process the raw video text descriptions obtained from the original dataset. As a result, approximately 1 million datasets requiring cleaning are identified. Subsequently, specific rules are designed based on this list to identify and eliminate any words of irrelevant information within the text. Finally, the cleaned texts are subjected to rewriting using the Vicuna-13B (Zheng et al., 2023), thereby ensuring its fluency and enhancing the overall quality.

**Interleaved Image and Text.** Multimodal-C4 (Zhu et al., 2023b) is used as interleaved image-text data in pretraining. Following OpenFlamingo (Awadalla et al., 2023), we filter images based on CLIP similarity score to ensure the relevance of the images and text in each document. Specifically, any image with a CLIP similarity score below the threshold of 0.32 for all text in the same document is discarded. From each document, we sample a random subsequence of L = 1024 and take up to the first N = 5 images included in the sampled sequence. This process results in long text with the inclusion of multiple images. Additionally, we randomly sample N = 5 images along with their corresponding sentences to construct a subsequence of L = 512. This approach yields N = 5 image-text pairs.

**Interleaved Video and Text.** Videos with interleaved subtitles text represent a valuable and scalable source of multimodal data that has received limited attention thus far. In our study, we introduced YT-Storyboard-1B dataset, which collected storyboard images from *YouTube*, utilizing the video-ids provided by the YT-Temporal-1B dataset, which encompasses a vast collection of 18 million videos, equating to a total of 1.8 billion storyboard images. Specifically, for each video, we crawl the storyboard images and subtitles files directly. Where the sampling time between storyboard images is fixed, so the start time of each image can be determined by the order. Subtitle files record the content of each subtitle, as well as the start and end times. Therefore, storyboard images and subtitles can be sorted according to their timestamps and adjacent subtitles can be merged to form an interleaved video-text sequence. By opting to collect storyboard images instead of raw video data, we eliminate the necessity of video decoding. Moreover, this approach leads to a substantial 20-fold reduction in data storage costs, resulting in increased download efficiency.

### A.1.2 TRAINING DETAILS

We report the detailed training hyperparameter settings of **Emu** during the pretraining in Table 5.

### A.1.3 ABLATIONS ON INTERLEAVED VIDEO-TEXT DATA

We conduct ablation experiments to study the effectiveness of the YT-Storyboard-1B dataset. Smaller model size, batch size and shorter training schedule are used compared to the final **Emu** model to reduce the training cost. The model consists of a large-sized EVA-CLIP (Sun et al., 2023) as visual encoder and LLaMA-7B (Touvron et al., 2023) as multimodal modeling module. Image-text and video-text pairs are used as common datasets, including LAION-COCO (lai, b), LAION-2B (Schuhmann et al., 2022) and WebVid-10M (Bain et al., 2021), with batch size of 256, 256, 32, respectively. YT-Storyboard-1B batch size of 32 is used. Models are trained for 7.2k gradient steps on image-text and video-text pairs and additional 2.4k gradient steps on YT-Storyboard-1B.

Table 5: Summary of pretraining hyperparameters of **Emu**.

| Configuration | **Emu** Pretraining |
|---|---|
| Vision encoder weight init. | EVA-CLIP (Sun et al., 2023) |
| Large language model weight init. | LLaMA-13B (Touvron et al., 2023) |
| Causal transformer weight init. | random init. |
| Vision encoder peak learning rate | 4e-5 |
| Large language model peak learning rate | 3e-5 |
| Causal transformer peak learning rate | 1e-4 |
| Warmup ratio | 0.2 |
| Learning rate schedule | cosine decay |
| Optimizer | AdamW (Loshchilov & Hutter, 2017) |
| Optimizer hyper-parameters | $\beta_1, \beta_2, \epsilon = 0.9, 0.98, 1e-6$ |
| Weight decay | 0.05 |
| Input image resolution | 224×224 |
| Iterations | 10k |
| Data | LAION-2B (Schuhmann et al., 2022), LAION-COCO (lai, b), MMC4 (Zhu et al., 2023b), Webvid-10M (Bain et al., 2021), YT-Storyboard-1B (Zellers et al., 2022) |
| Batch size per dataset | 128, 128, 64, 16, 16 |

Table 6: Quantitative comparison of w/wo interleaved video and text data during pre-training. $k$ is the number of in-context examples, and we used the same example selection approach (*i.e.* RICES (Yang et al., 2022b) ) as Flamingo (Alayrac et al., 2022). * indicates that the zero-shot prompt is built by using two examples from the task, where their corresponding images have been removed.

| Setting | COCO | OKVQA | | MSVDQA | | MSRVTTQA | |
|---|---|---|---|---|---|---|---|
| | $k$=0 | $k$=0 * | $k$=4 | $k$=0 * | $k$=4 | $k$=0 * | $k$=4 |
| Emu-7B w/o YT-Storyboard-1B | 110.8 | **43.9** | 44.6 | 30.2 | 31.1 | 16.9 | 19.9 |
| Emu-7B w/ YT-Storyboard-1B | **112.9** | 42.3 | **45.7** | **30.8** | **34.9** | **17.9** | **20.8** |

With YT-Storyboard-1B incorporated in the pretraining stage, Emu-7B achieves better zero-shot performance on MS-COCO (Chen et al., 2015), MSVDQA (Xu et al., 2017) and MSRVTTQA (Xu et al., 2017). Besides, YT-Storyboard-1B also brings stronger in-context learning capability under 4-shot evaluation.

## A.2 VISUAL DECODING

### A.2.1 DATASET DETAILS

LAION-Aesthetics (lai, a) is the subset of LAION-5B (Schuhmann et al., 2022) which have relatively high aesthetics quality while LAION-COCO (lai, b) has relatively high image-text correlation. To empower the visual decoder to possess the ability of decoding visual embeddings with both high quality and high relevance to text prompts, we use both LAION-COCO and LAION-Aesthetics for visual decoding training. More specifically, we filter all text prompts with length greater than 150 to preserve a large enough batch size and prevent the GPU memory overflow. This rule discards about 8% of LAION-Aesthetics and 0.01% of LAION-COCO data, which has little effect on data diversity.

### A.2.2 TRAINING DETAILS

The detailed training setups are listed in Table 7.

Table 7: Summary of **Emu** visual decoder training hyperparameters.

| Configuration | Visual Decoder |
|---|---|
| Weight init | Stable Diffusion v1.5 |
| Batch size | $50 \times 4 \times 8$ |
| Iterations | 15k |
| Learning rate | warm up to 1e-4 for the first 5k, decrease to 5e-5 and 1e-5 at 10k and 14k |
| Input image resolution | $512 \times 512$ |
| Objective | $\epsilon$-prediction |
| Optimizer | AdamW (Loshchilov & Hutter, 2017) |
| Optimizer hyper-parameters | $\beta_1, \beta_2, \epsilon = 0.9, 0.999, 1e-8$ |
| Weight decay | 1e-2 |
| Data | LAION-COCO (lai, b), LAION-Aesthetics (lai, a) |
| Data ratio | 7:2 |

## B INSTRUCTION TUNING

### B.1 DATASET AND TRAINING DETAILS

We collect publicly available language, image and video instruction datasets for instruction tuning.

- Language instructions: ShareGPT contains about 70K user dialogues with ChatGPT or GPT-4, and Alpaca (Taori et al., 2023) dataset contains 52K instruction-following data generated using self-instruct (Wang et al., 2022c) from OpenAI's `text-davinci-003`.

- Image instructions: we use LLaVA (Liu et al., 2023b) dataset consisting of three types of visual instructions, conversation, detailed description, and complex reasoning, with a total number of 158K image-text instruction-following samples. In our preliminary experiments, we found the instruction-tuned model often generates instruction-irrelevant detailed descriptions of the image. Thus, we remove the detailed description subset of LLaVA. We also find a bad pattern 'on top of the back of' in the model's response, and we filter all data that contains this pattern. The resulting 130K LLaVA subset is used for instruction tuning.

- Video instructions: we use VideoChat-11K (Li et al., 2023c) and a subset of Video-ChatGPT-100k (Maaz et al., 2023) as our video-instruction dataset. VideoChat-11K dataset is built from WebVid-10M consisting of 7K detailed video descriptions and 4K video conversations. Video-ChatGPT-100k is built from ActivityNet, and we sample an around 30K subset that includes only videos under one minute.

We use a batch size of 128 and train for 10K steps, with 3 epoches for ShareGPT, Alpaca and LLaVA datasets, and 60K samples for video-instruction data. The learning rate linearly warms up to 1e-5 in the first 500 steps, then decays to zero with a cosine schedule. The overall instruction tuning stage takes around 16 hours with 16 A100-80G GPUs. We attach LoRAs (Hu et al., 2022) on all linear projections of the self-attention layer, with the LoRA rank and alpha being 16.

### B.2 SYSTEM MESSAGES

We use different system messages for language-instruction, image-instruction and video-instruction datasets, as shown in Table 8.

Table 8: Summary of the prompting template.

| Task Type | <System Message> |
|---|---|
| Language Instruction Datasets | None |
| Image Instruction Datasets | You are a helpful assistant and you will be presented with an image: [IMG]ImageContent[/IMG]. You will be able to see the image after I provide it to you. Please answer my questions based on the given image. |
| Video Instruction Datasets | You are a helpful assistant and you will be presented with a video consisting of multiple chronological images: [IMG]ImageContent[/IMG]. You will be able to see the video after I provide it to you. Please answer my questions based on the given video. |

Table 9: Summary of the evaluation benchmarks.

| | Dataset | Task | Split | Metric |
|---|---|---|---|---|
| Image | COCO Text2Image | Text-to-Image Generation | Val | FID($\downarrow$) |
| | COCO Caption | Scene description | Test | CIDEr($\uparrow$) |
| | VQAv2 | Scene understanding QA | Test-dev | VQA acc.($\uparrow$) |
| | OKVQA | External knowledge QA | Val | VQA acc.($\uparrow$) |
| | VizWiz | Scene understanding QA | Test-dev | VQA acc.($\uparrow$) |
| | VisDial | Image Dialogue | Val | NDCG($\uparrow$) |
| Video | MSVDQA | Event understanding QA | Test | Top-1 acc.($\uparrow$) |
| | MSRVTTQA | Event understanding QA | Test | Top-1 acc.($\uparrow$) |
| | NextQA | Temporal/Causal QA | Test | WUPS($\uparrow$) |

## C EVALUATION

### C.1 BENCHMARKS

**Emu** excels at performing diverse types of completion in multimodal sequences by accepting multimodal prompts, including text, images, videos, or their combinations, and generating comprehensive multimodal responses. To evaluate the capabilities of **Emu**, we conduct extensive benchmark tests covering various tasks, which are summarized in Table 9. Specifically, we meticulously select 9 benchmarks that encompass multimodal image/video and language tasks, including text-to-image generation, visual question answering for both images and videos, and image-based visual dialogue. When benchmarking OKVQA, we use VQAv2 evaluation code[3] and stem the answers using Porter stemming to consolidate answers following Marino et al. (2019). For other tasks, we either submit our results for evaluation on the official website or use standard evaluation code.

### C.2 ZERO-SHOT EVALUATION

**Prompt Template.** To ensure that the model outputs answers in the required style for the benchmark tests, we prompt **Emu** and **Emu-I** with task-specific templates, as shown in Table 10. For each type of task, we have developed dedicated templates to structure the model's output. In these templates, "{question}" will be replaced with the question from the question-answering task, "{history

---

[3]https://github.com/GT-Vision-Lab/VQA

`question}`" will be replaced with the historical question from the multi-turn visual dialogues, and similarly "history answer" will be replaced with the historical annotated answer from the multi-turn visual dialogues. Then, the image/video will be added before the text as input to the model. Additionally, we implement post-processing techniques to filter out commonly occurring redundant phrases such as "it is", "it's", "a", "an", and "the". Furthermore, the model is required to output "unanswerable" for questions that cannot be answered in the VizWiz dataset. To achieve this, we augment the template by adding the phrase "`is the answer known?`" and prompt the model to respond with either "yes" or "no" by constraining the model generation. If the model responds with "no", we immediately return the answer as "unanswerable". On the other hand, if the model responds with "yes", we proceed to prompt the model to provide a valid answer.

**Multimodal Chain-of-Thought Prompting.** To enhance the capabilities of the pretrained model, we utilize the Multimodal Chain-of-Thought prompting technique. Initially, when presented with an image or video, we employ a prompt to guide the model in generating a descriptive caption. Subsequently, the model is given both the caption and a task-specific prompt to generate the final result. The complete prompt template is shown in Table 10, where the "`{caption}`" tag in template will be replaced with the descriptive text generated by **Emu**. The experimental results demonstrate that this test-time technique effectively improves the model's performance without any additional data, leveraging the inherent capability of the model itself.

**Text-only Examples Prompting.** To ensure a fair comparison with Flamingo, we include results obtained through text-only examples prompting, denoted by an asterisk (*) in Table 1. We adopt the same approach as Flamingo in selecting examples (*i.e., RICES*). This involves utilizing two text-only examples from the task as prompts, without any accompanying images (similar to the few-shot text prompts). During the evaluation process, we observed that this approach effectively formats the model's output, regardless of the label format of the datasets and the evaluation metrics employed, enabling a more accurate reflection of its true performance.

### C.3 FEW-SHOT EVALUATION

In the few-shot evaluation settings, we incorporate a few example samples as prefixes in the template and connected the few-shot examples using ".    ". Additionally, like Flamingo, we employ the Retrieval In-Context Example Selection (RICES) approach to select the few-shot examples.

To implement RICES, we begin by randomly selecting 5000 training set samples for each dataset. Then, using the pretrained EVA-CLIP model, we extract features from both the training set images/videos and the test set images/videos. For each test set sample, we select examples from the training set based on the highest cosine similarity using the extracted features, including them in the prompt. For the video-text task, we retrieve similar videos from the training set by comparing the mean of frame-level visual features extracted from our pretrained EVA-CLIP model.

Furthermore, we discover that the support video examples didn't require too many frames, which could exceed the LLM's context length limit. Therefore, we sample 8 frames for the given video and only 2 frames for the corresponding support video examples.

### D COMPARISON WITH RELATED WORK

The results are presented in Table 11, where **Emu** achieves state-of-the-art results on 5 out of 6 benchmarks evaluated.

CM3Leon Yu et al. (2023a) has similar motivation with us to train a unified image-to-text and text-to-image model. The most significant difference lies in that CM3Leon discretizes images, but we directly input and output image continuous features. We can find that CM3Leon performs much worse than **Emu** on image-to-text tasks.

AnyMAL Moon et al. (2023) is a large-scale Multimodal LLM that can process any modality, but have relatively weak performance on image to text tasks.

Table 10: Summary of the prompting template.

| Model | Type | Template |
|-------|------|----------|
| **Emu** | Image Captioning | `describing the image in detail. the image shows` |
| | Image QA | `a picture of {caption}. based on the picture, {question} short answer:` |
| | Image Dialog | `a picture of {caption}. based on the picture, {history question} short answer: {history answer}. ... based on the picture, {question} short answer:` |
| | Video Event understanding QA | `a video of {caption}. a question about the video: {question} answer:` |
| | Video Temporal/Causal QA | `a video of {caption}. a question about the video: {question} short answer:` |
| **Emu-I** | Image Captioning | `[USER]: please provide an accurate and concise description of the given image. [ASSISTANT]: the image depicts a photo of` |
| | Image QA | `[USER]: based on the content of the image and common sense, please provide an accurate answer consisting of only one word or phrase. [ASSISTANT]: the answer is:` |
| | Image Dialog | `[USER]: {history question} [ASSISTANT]: {history answer}.<\s> ... [USER]: {question} [ASSISTANT]:` |
| | Video Event understanding QA | `[USER]: based on the content of the video and common sense, please provide an accurate answer consisting of only one word or phrase. {question} [ASSISTANT]: the answer is:` |
| | Video Temporal/Causal QA | `[USER]: based on the content of the video and common sense, please provide an accurate short answer. {question} [ASSISTANT]: the answer is:` |

Table 11: Zero-shot comparison with concurrent work.

| Models | COCO (Image2Text) | COCO (Text2Image) | VQAv2 | OKVQA | VizWiz | VisDial |
|--------|-------------------|-------------------|-------|-------|--------|---------|
| CM3Leon | 61.6 | **10.82** | 47.6 | 23.8 | 37.6 | 22.6 |
| AnyMAL-13B | 99.5 | - | 59.6 | 33.1 | 24.4 | - |
| **Emu** | 112.4 | 11.66 | 52.0 | 38.2 | 34.2 | 47.4 |
| **Emu** * | - | - | 52.9 | 42.8 | 34.4 | 47.8 |
| **Emu-I** | **120.4** | - | 57.2 | 43.4 | 32.2 | 43.0 |
| **Emu-I** * | - | - | **62.0** | **49.2** | **38.3** | **51.1** |

# E   QUALITATIVE CASES

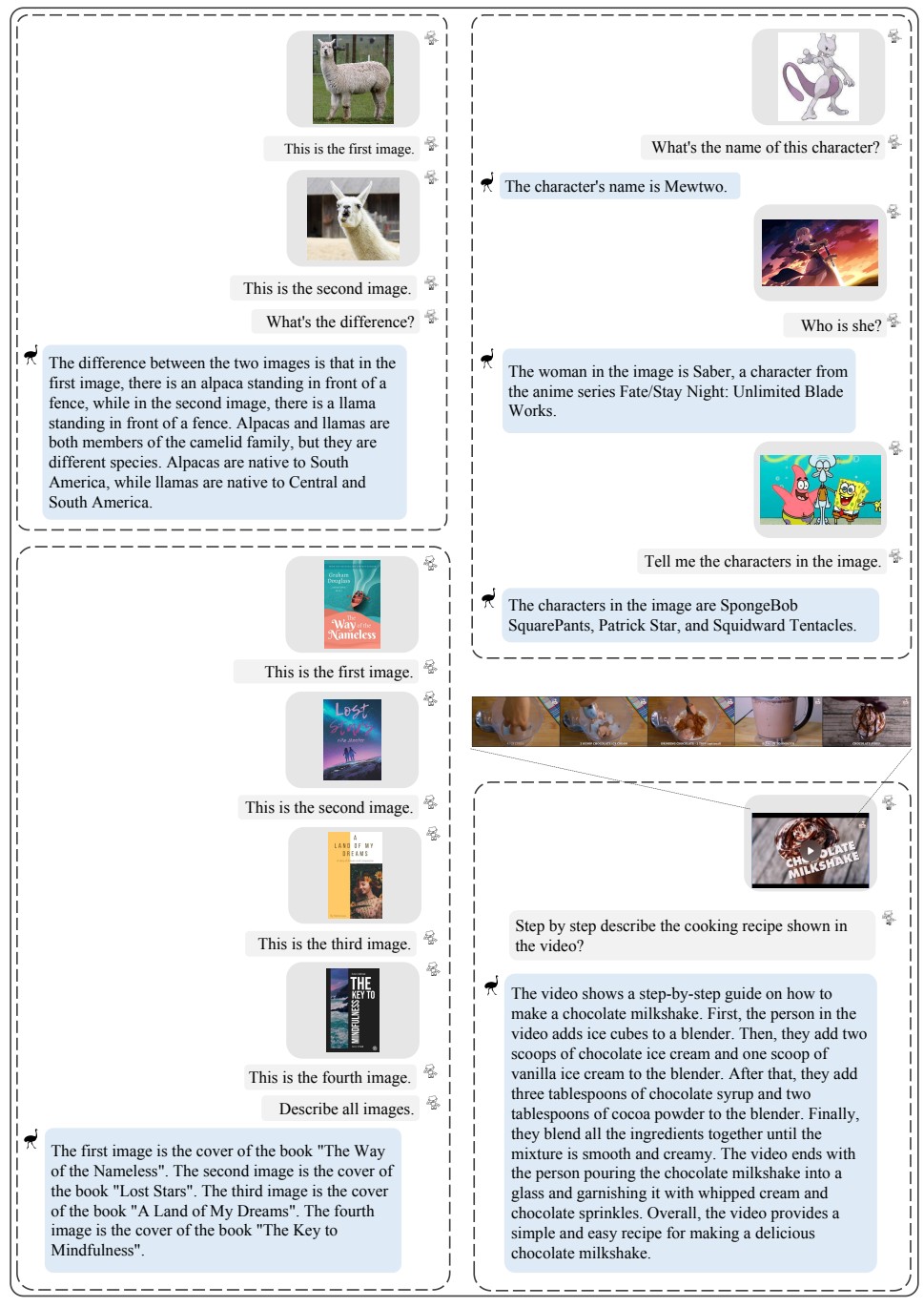

Figure 4: Examples of interleaved multi-image understanding(left side), real-world knowledge grounding(upper right), detailed video understanding(lower right).

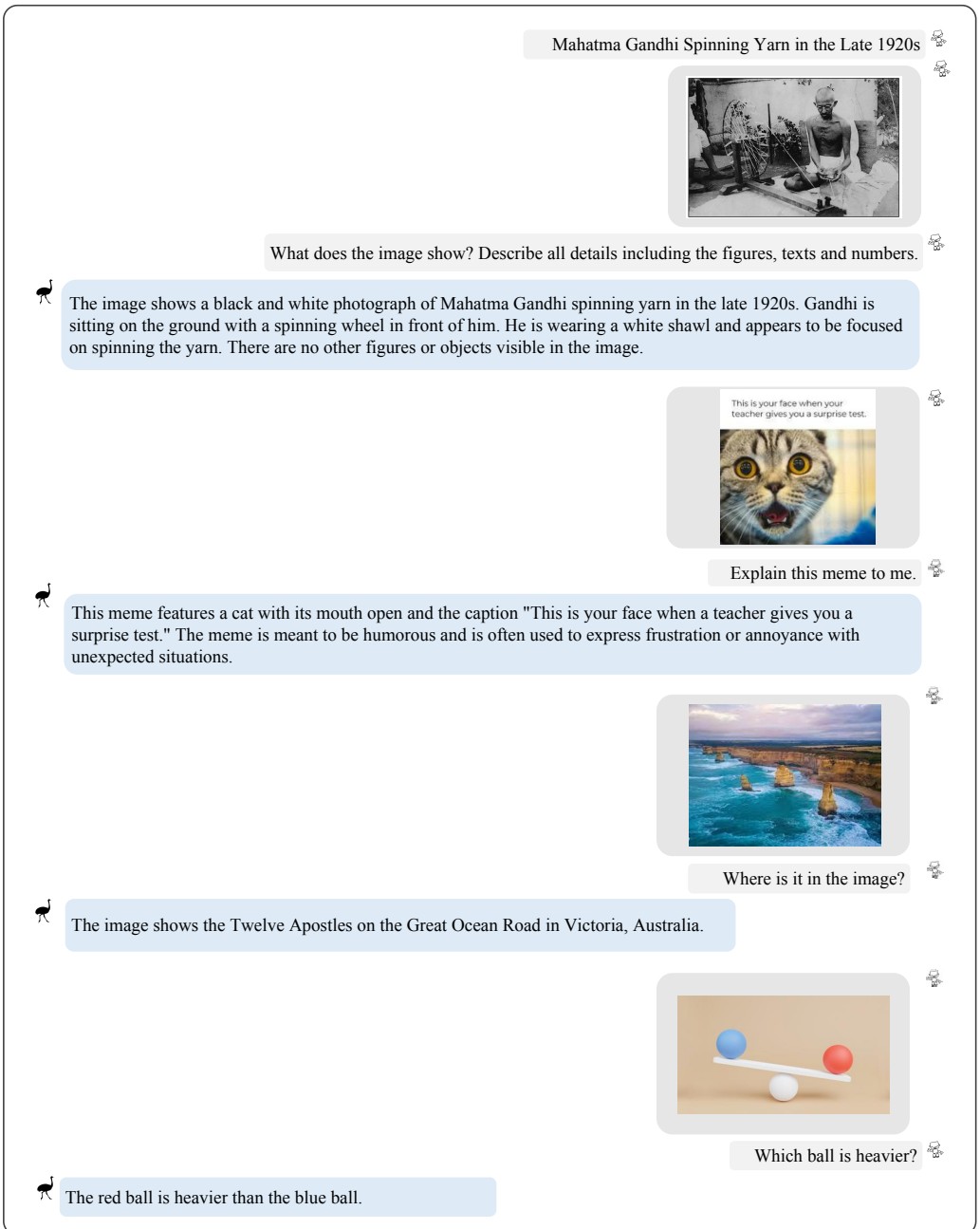

Figure 5: Examples of multimodal assistant.

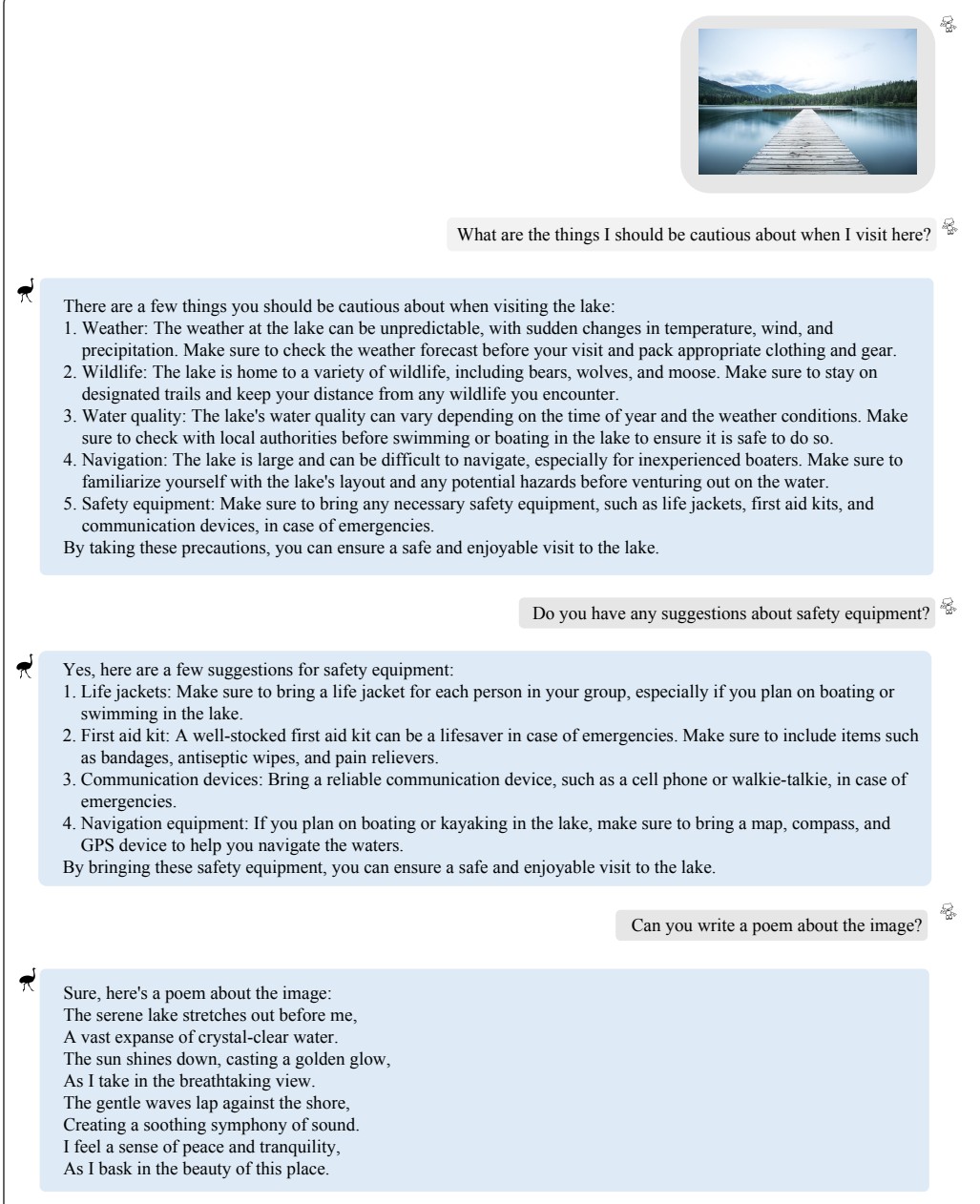

Figure 6: Examples of multi-turn dialogue.

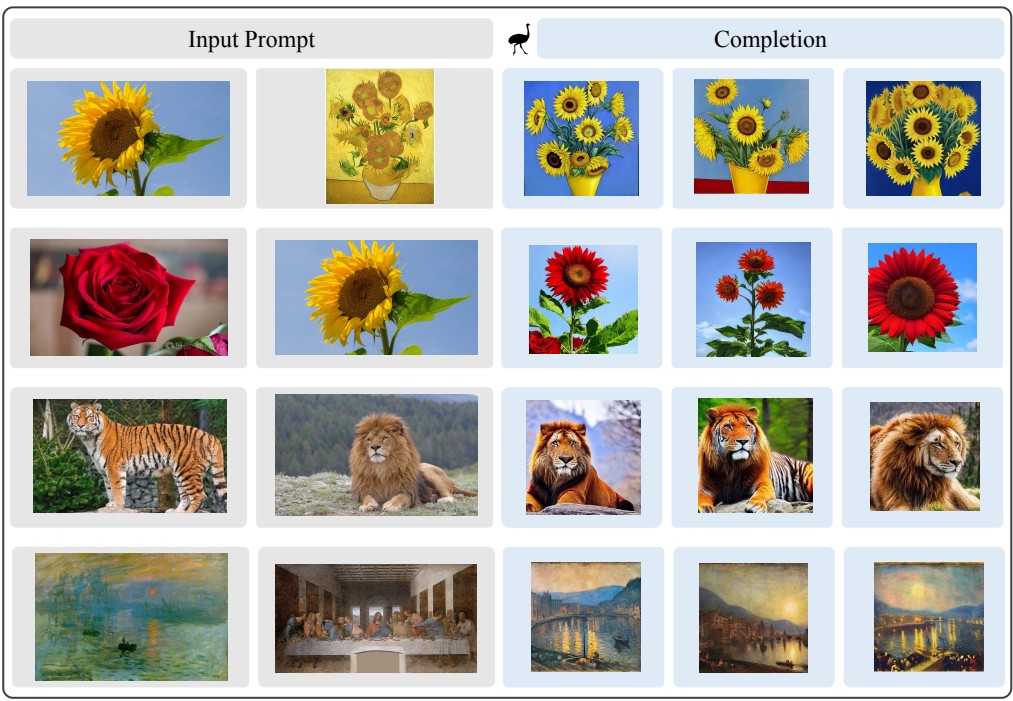

Figure 7: Examples of image blending.

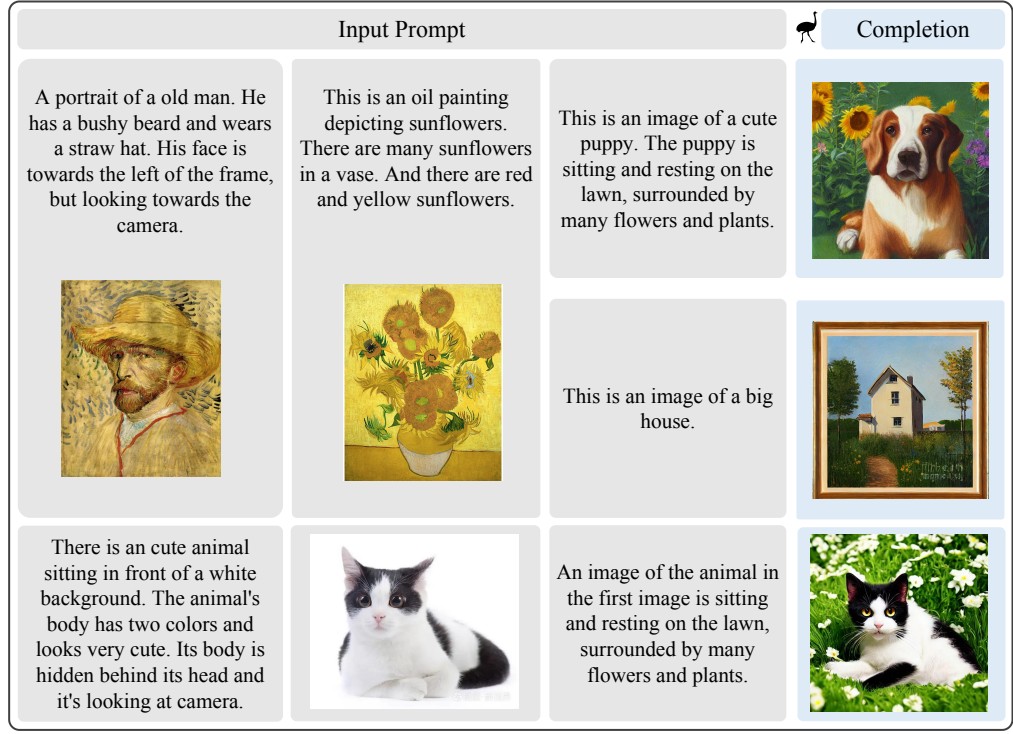

Figure 8: Examples of in-context text-to-image generation.

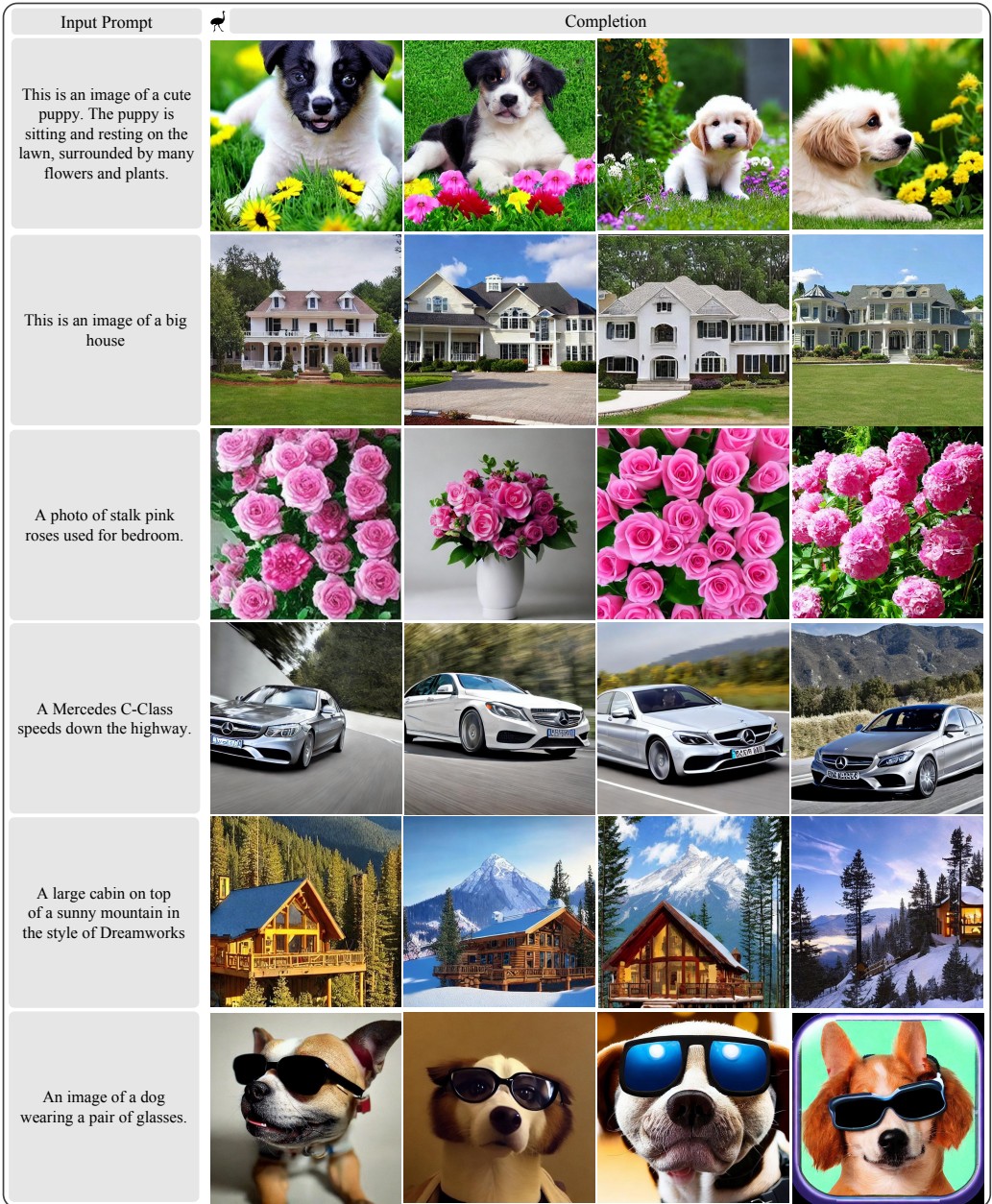

Figure 9: Examples of text-to-image generation.

Input Image

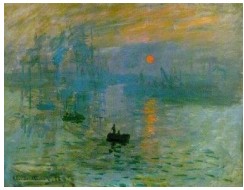

- - - - - - - - - - - - - - - - - - - - - - - - - - - - - - - - - - - - - - - - - - - - - - - - - - - - - -

Input Text          Describe the image in detail.

- - - - - - - - - - - - - - - - - - - - - - - - - - - - - - - - - - - - - - - - - - - - - - - - - - - - - -

Model Output

Emu

The image is a painting by Claude Monet titled "Impression, Sunrise". It depicts a serene scene of the sun rising over the water. The painting is done in a loose, impressionistic style, with visible brushstrokes and a focus on capturing the essence of the scene rather than the details. The colors are muted and soft, with shades of blue, pink, and yellow dominating the canvas.

In the foreground of the painting, there is a small boat bobbing up and down on the water. The sun is just peeking over the horizon, casting a warm, golden light over the scene. The water is calm and still, reflecting the colors of the sky above.

In the background of the painting, there are a few buildings visible, but they are mostly obscured by the mist and haze of the early morning. The sky is a mix of pink and blue, with a few wispy clouds drifting lazily across the sky.

Overall, the painting is a beautiful and serene depiction of a peaceful morning on the water. The soft colors and loose brushstrokes evoke a sense of calm and tranquility, inviting the viewer to take a moment to appreciate the beauty of nature.

LLaVA

The image is a captivating scene of a small boat floating on a body of water, possibly a bay or a river, during a beautiful sunset. The boat is positioned towards the center of the picture, surrounded by a serene atmosphere.

There are several other boats scattered throughout the image, some closer to the edges of the frame, while others are positioned more towards the center. The presence of multiple boats adds a sense of depth and interest to the scene.

The sunset casts a warm glow on the water and the boats, creating a picturesque and tranquil environment for those on the small boats to enjoy the view.

mPLUG-Owl

The painting depicts a beautiful sunrise over the ocean, creating a stunning and serene atmosphere. The scene is set against a backdrop of a blue-gray sky, with a bright yellow sun peeking out from behind a cloud.

The water in the foreground of the painting is calm and peaceful, gently reflecting the sun's rays. Several sailboats can be seen floating in various positions, adding to the tranquil atmosphere of this early morning moment. In addition to boats, there are several birds flying in different directions, enjoying the beautiful view.

InstructBLIP

In this painting, Claude Monet's famous impressionist artwork titled "Sunrise" takes center stage. The painting depicts the beautiful sunrise over the horizon, with its warm orange and yellow hues spreading across the sky. The water is also a significant part of the painting, with several boats sailing along it. The sunrise is captured in the middle of the painting, with boats scattered across the scene. In the background, the boats and the water are surrounded by various colors, with some areas featuring a lighter, brighter tone. The sunrise creates a serene and peaceful atmosphere, while the boats add a touch of activity to the scene.

Figure 10: Comparison of **Emu** with other methods on **image captioning task**.

**Input Image**

---

**Input Text**   Tell me 8 books written by the woman and recommend one to me.

---

**Model Output**

**Emu**

Here are 8 books written by Agatha Christie:

1. "The Mysterious Affair at Styles" (1920)
2. "The Secret Adversary" (1922)
3. "The Murder on the Links" (1923)
4. "The Man in the Brown Suit" (1924)
5. "The Secret of Chimneys" (1925)
6. "The Murder of Roger Ackroyd" (1926)
7. "The Big Four" (1927)
8. "The Murder at the Vicarage" (1930)
Recommendation: "The Murder of Roger Ackroyd" (1926)

"The Murder of Roger Ackroyd" is considered by many to be Agatha Christie's greatest mystery novel. It is a classic whodunit with a twist ending that will keep you guessing until the very end.

**LLaVA**

The woman in the image, Agatha Christie, has written numerous books, totaling up to 66 novels, including mysteries and thrillers. One of her most famous and highly recommended books is "And Then There Were None." This book is a classic whodunit novel that tells the story of ten strangers who are lured to an isolated island and then murdered one by one, according to the rules of the famous nursery rhyme. The intricate plot, unique setting, and engaging characters make it an excellent read for anyone who enjoys mysteries and suspense.

**mPLUG-Owl**

Agatha Christie was a prolific British author who wrote over 80 novels, short stories, and plays. Some of her most famous works include:

1. Murder on the Orient Express (1934): A classic mystery novel about the investigation of a murder on a luxury train.
2. And Then There Were None (also known as Ten Little Indians) ( 10 Little Indian, 2015)
3. The Murders on The Rue Morgue (first published in 90s) - A detective story about a series of murders in Paris.
4. A MurDER IS ANNOUNCED (published in the 40's). A play that was adapted for a movie and later became a TV mini-series. It'll give you a glimpse into her writing style.

**InstructBLIP**   the spiderwick chronicles

Figure 11: Comparison of **Emu** with other methods in term of **following human instructions**.

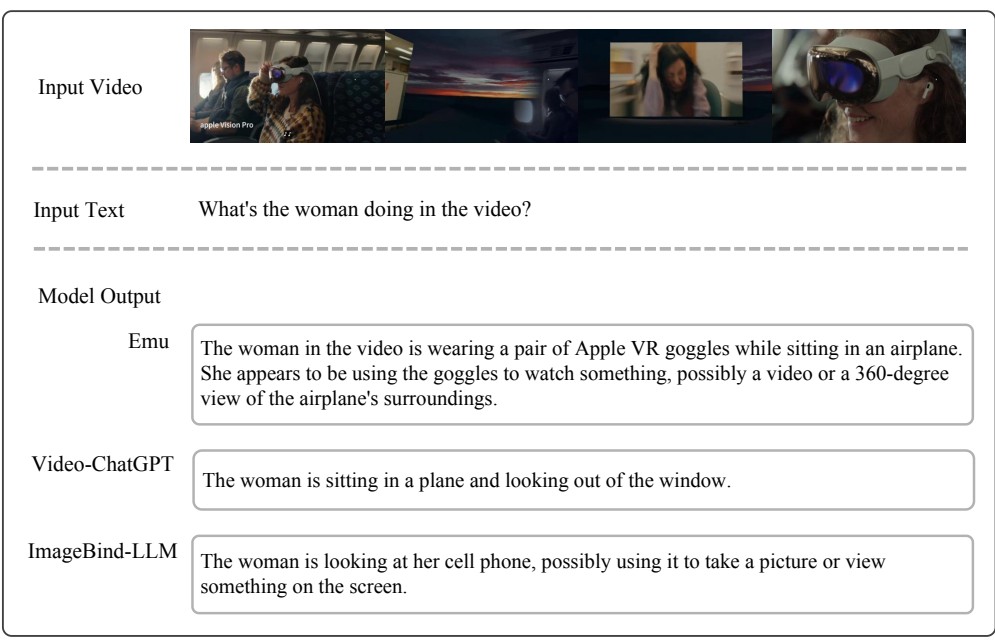

Figure 12: Comparison of **Emu** with other methods in term of **following human instructions**.

