# OpenReview forum: "Emu: Generative Pretraining in Multimodality"
_ICLR.cc/2024/Conference — ICLR 2024 poster_

### Official Review · Reviewer_paGX · 2023-10-29

**Soundness:** 3 good
**Presentation:** 3 good
**Contribution:** 3 good
**Rating:** 6
**Confidence:** 4

**Summary:**

This paper presents Emu, a foundation model designed for multimodal tasks, capable of handling text, image, and video inputs. It uses a unified autoregressive training approach to encode visual signals and text tokens into an interleaved sequence. Trained on diverse data sources, Emu is versatile in both image-to-text and text-to-image tasks, and performs well in zero-shot and few-shot inference. It also shows promise in extended applications like multimodal assistants, using instruction tuning.

**Strengths:**

- It is a strong paper that investigates a unified multimodal pre-training recipe on creating generative model that inputs and produces interleaved image and text sequence, capable of multimodal understanding and image generation.
- The model is comprehensively evaluated, on a wide variety of tasks, both generative and discriminative.

**Weaknesses:**

- The comparison of image-text and video-text tasks should be also made with other state-of-the-art multimodal models, such as GPT-4V, PaLI-X (55B). It is okay to underperform given that those models are (likely) larger in the model size. The important part is to give reader a full picture of how this work compares to those important (private) models

**Questions:**

- I am curious if the pre-trained model can do more complex tasks such as the in-context subject-driven generation introduced in the paper [1]. Given that the model is general enough to handle prompts such as "Given the a few images about a common subject {subject_name}: {image_1}, {image_2}, {image_3}. Generate a new rendition of this same subject, in the Hall of Mirrors in Versailles.". If would be amazing if such complex task can be done in zero-shot.

[1] Subject-driven text-to-image generation via apprenticeship learning

**Details Of Ethics Concerns:**

This paper does not evaluate, analyze or discuss the potential bias the model could have learned from the data. Particularly given that it is also an image generation model, I am wondering if language bias could be transferred to visual domain. For example, we know that in NLP,  programmer is often higher correlated to male, and I wonder if such situation would also transfer to the text-to-image generation. If you prompt the model to generate 'a programmer standing in front of Googleplex', how often would it be male vs. female?

---

> ### Author Response · Authors · 2023-11-20
> **Response to Reviewer paGX**
>
> Thank the reviewer for the comments. We have uploaded a revision of our paper. Below we address the detailed questions.
>
> **Q1: Comparision to PALI-X and GPT-4V**
>
> We have added PaLI-X's results in our paper for comparison. We would like to clarify that the comparison may be unfair as PaLI-X is 1) much larger, 2) pretrained in-house data, 3) fine-tuned on each academic dataset rather than zero-shot. As for GPT-4V, we are unable to evaluate it because its API is just released several days ago and we have limited access.
>
> **Q2: Complex tasks like subject-driven image generation**
>
> Thanks for the great suggestion. This is a challenging problem and the existing works use specialist architectures or specialist models for each personized subject. We show that our model can perform this task in some cases (as shown in the 4th row of Figure 1), but it is too hard for the current model to solve this problem proficiently. However, we think this is an interesting problem and we believe that after further scaling up the data and model sizes, this kind of zero-shot ability might emerge, given the flexibility of our pretraining framework.
>
> **Q3: Ethics Concerns**
>
> We thank the reviewer for the advice. We have included a new page in our paper to discuss ethics concerns and possible mitigation strategies in our paper.

---

> ### Author Response · Authors · 2023-11-22
> **Anticipating Your Participation as Reviewer-Author Discussion Deadline Approaching**
>
> We express our sincere gratitude to the reviewer for dedicating time to review our paper. **We have provided comprehensive responses to all the questions including 1) comparison to PALI-X and GPT-4V, 2) subject-driven image generation, and 3) ethics concerns.** As the discussion deadline looms within a day, we would like to inquire if our responses have adequately addressed your questions. We anticipate your participation in the Reviewer-Author discussion phase, as your insights are invaluable for refining and enhancing the quality of our paper. Should you have any additional queries or require further clarification, please do not hesitate to inform us. We are more than willing to address any concerns and ensure a comprehensive resolution. Thank you for your time and consideration.

---

### Official Review · Reviewer_qbfH · 2023-10-30

**Soundness:** 3 good
**Presentation:** 3 good
**Contribution:** 3 good
**Rating:** 6
**Confidence:** 4

**Summary:**

The paper presents a novel multimodal foundation model named Emu, designed to handle both images and text in a multimodal context. The primary innovation lies in its autoregressive training process which can accommodate single-modality or multimodal data inputs like images, text, and video. Visual signals are initially encoded into embeddings, which, along with text tokens, form an interleaved input sequence. Emu is trained end-to-end with a unified objective which alternates between classifying the next text token and regressing the next visual embedding within the multimodal sequence. This model's flexibility allows it to harness diverse pretraining data sources, including videos with interleaved frames and text, webpages with interleaved images and text, as well as web-scale image-text pairs and video-text pairs. By doing so, it demonstrates superior performance across a range of zero-shot and few-shot tasks like image captioning, visual question answering, and text-to-image generation.

Emu's architecture is divided into four components: the Visual Encoder, Causal Transformer, Multimodal Modeling, and Visual Decoder. The Visual Encoder and Causal Transformer handle the conversion of images and videos into a format compatible with the Multimodal Modeling LLM (Large Language Model). Specifically, the visual data is first encoded into dense visual features, which are then transformed into a fixed number of visual causal embeddings. These visual causal embeddings, along with text tokens, are fed into the Multimodal Modeling LLM, which performs unified autoregressive modeling. Post-training, a Visual Decoder is fine-tuned to convert visual embeddings back into realistic images. The training data spans web-scale collections of image-text pairs, video-text pairs, interleaved image-text data, and interleaved video-text data, with Emu being pretrained on these multimodal data sequences under a unified objective of predicting the next element in a multimodal sequence.

The evaluation of Emu covered a broad spectrum of vision-language tasks, demonstrating its effectiveness and superior performance in comparison to other large multimodal models in many instances. In zero-shot evaluations, Emu showcased remarkable results in tasks like COCO captioning and VizWiz VQA. In few-shot evaluations, Emu continued to exhibit strong performance across various image and video question answering tasks. The qualitative evaluation also underlined Emu's real-world knowledge grounding, detailed video understanding, and in-context text-to-image generation capabilities. Moreover, the paper also explores an "in-the-wild" evaluation and instruction tuning to align Emu with human instructions further. Through its innovative architecture and comprehensive evaluation, Emu lays down a significant milestone in the journey towards more capable and versatile multimodal AI models.

**Strengths:**

- The paper is well-organized and the problem is clearly defined. The authors provide a comprehensive introduction to the problem and the proposed solution, Emu, which appears to be novel and well thought out.

- The unified objective for both text and visual data seems to be a promising approach to handle multimodal tasks, and the autoregressive training process is well justified.

- The authors have undertaken extensive evaluations including zero-shot, few-shot, and in-the-wild evaluations, showcasing the model's performance across a variety of tasks. The performance of Emu, particularly in zero-shot tasks, appears to be impressive and competitive with other state-of-the-art models.

- The qualitative evaluation provided a good insight into the model's capabilities in real-world scenarios, which is a strong point of this paper. It’s interesting to see the model's performance in text-to-image generation, image blending, and real-world knowledge grounding.

- Incorporation of various pre-training data sources, including videos with subtitles and image-text interleaved data, could contribute to the model's strong performance and is a positive aspect of this work.

**Weaknesses:**

- The paper misses some key and very relevant comparative works like Cm3Leon (https://arxiv.org/abs/2309.02591), AnyMAL(https://arxiv.org/abs/2309.16058) etc. These papers should be compared against and explained how the authors work differes from the same.
- It would be beneficial to see a discussion on the scalability of Emu with respect to the size and diversity of training data, and how the model might perform with fewer resources or less diverse data.
- Autoregressive models are know to be notoriously slow at inference time due to the sequential nature of their execution. A comparative study of the same and ideas to make it better could help the paper.
- The image generation models are capable of generating harmful content like pornography, child abuse etc. These also have human faces which are not in compliance with privacy laws in several states and countries and the authors dont talk about taking any steps to ensure compliance to the same. This should potentially be addressed to ensure the model isnt misused.

**Questions:**

- The paper could benefit from a deeper discussion on the limitations of Emu, and potential strategies for overcoming these limitations in future work. Maybe the authors can add the same?

**Details Of Ethics Concerns:**

The image generation models are capable of generating harmful content like pornography, child abuse etc. These also have human faces which are not in compliance with privacy laws in several states and countries and the authors dont talk about taking any steps to ensure compliance to the same. This should potentially be addressed to ensure the model isnt misused.

---

> ### Author Response · Authors · 2023-11-20
> **Response to Reviewer qbfH**
>
> We thank the reviewer for the helpful comments. We have uploaded a revision of our paper. Below we address the detailed questions.
>
> **Q1: Missed baselines.**
>
> We thank the reviewer for providing the reference, and we included the performance comparison to these baselines below. We have updated a discussion and comparison in the paper (Section D). It is essential to note that the mentioned papers were released to arXiv less than one month before the ICLR submission deadline, e.g., Sept 5 for Cm3Leon, Sept 27 for AnyMAL, and Sept 28 for ICLR DDL, and our project was concluded prior to our awareness of these concurrent papers.
>
> |                   | COCO FID  | COCO Caption | VQAv2    | OKVQA    | VizWiz   | VisDial  |
> | ----------------- | --------- | ------------ | -------- | -------- | -------- | -------- |
> | CM3Leon (Sep. 5) | **10.82** | 61.6         | 47.6     | 23.8     | 37.6 | 22.6     |
> | AnyMAL-13B (Sep. 27) | - | 99.5         | 59.6     | 33.1     | 24.4 | -     |
> | Emu               | 11.66     | 112.4    | 52.0 | 38.2 | 34.2     | 47.4 |
> | Emu *               | -    | -    | 52.9 | 42.8 | 34.4  | 47.8 |
> | Emu-I               | -     | **120.4**    | 57.2 | 43.4 | 32.2     | 43.0 |
> | Emu-I *               |  -  | -    | **62.0** | **49.2** | **38.3**   | **51.1** |
>
> **Q2 Scalability of Emu w.r.t. the size and diversity of training data.**
>
> We have added the experiments of smaller scale and less diverse data.
>
> |                      | COCO   | OKVQA  | MSVDQA* |
> | -------------------- | ------ | ------ | ------- |
> |                      | 0-shot | 0-shot | 0-shot  |
> | Emu-7B w/o YT-SB-1B  | 110.8  | 27.6   | 30.2    |
> | Emu-7B w. YT-SB-1B   | 112.9  | 31.3   | 30.8    |
> | Emu-14B w. YT-SB-1B  | 112.4  | 38.2   | 34.3    |
> | Emu-14B w. 50k steps | **120.5**  | **41.6**   | **36.0**    |
>
>  *the zero-shot prompt of MSVDQA is built by using two examples from the task
>
> With scaling up the training data, i.e., the inclusion of additional data from YT-SB-1B, 7 billion parameters Emu demonstrates a noteworthy improvement in zero-shot performance on tasks such as OKVQA(+3.7), and MSVDQA(+0.6).  With an increase in model size to 14 billion parameters, zero-shot performance experience further enhancement. Notably, zero-shot performance gains +6.9 on OKVQA, and +3.5 on MSVDQA. This emphasizes the positive impact of scalability, considering both the size and diversity of training data, along with the increased model capacity, on the overall performance across a diverse set of tasks.
>
> To further show the scalability of Emu, we scale up the  training schedule with more iterations and training samples, zero-shot performance gains +8.1 on COCO caption, +3.4 on OKVQA, and +1.7 on MSVDQA.
>
> **Q3: Slow inference speed of autoregressive models.**
>
> We agree that autoregressive model has its unique problems. However, we want to note that autoregressive model is the predominant architecture now and a lot of work to improve the inference speed have been devised, e.g. Flash Decoding. And these techniques also work in our case. Improving efficiency or speed is out of the scope of this paper, instead we offer a new perspective and approach to enable generative pretraining in multimodality. Please let us know if we misunderstood the question.
>
> **Q4: Limitations of Emu**
>
> Thank the reviewer for the suggestion. We have included a section (Section 7) in our updated paper to discuss the limitations of Emu and potential future work.
>
> **Q5: Ethics Concerns**
>
> We thank the reviewer for the advice. We have included a new page in our paper to discuss ethics concerns and possible mitigation strategies in our paper.

---

> ### Author Response · Authors · 2023-11-22
> **Anticipating Your Participation as Reviewer-Author Discussion Deadline Approaching**
>
> We express our sincere gratitude to the reviewer for dedicating time to review our paper. **We have provided comprehensive responses to all the questions including 1) missed baselines, 2) scalability, 3) limitations, and 4) ethics concerns.** As the discussion deadline looms within a day, we would like to inquire if our responses have adequately addressed your questions. We anticipate your participation in the Reviewer-Author discussion phase, as your insights are invaluable for refining and enhancing the quality of our paper. Should you have any additional queries or require further clarification, please do not hesitate to inform us. We are more than willing to address any concerns and ensure a comprehensive resolution. Thank you for your time and consideration.

---

### Official Review · Reviewer_qjEg · 2023-11-01

**Soundness:** 3 good
**Presentation:** 3 good
**Contribution:** 3 good
**Rating:** 6
**Confidence:** 4

**Summary:**

The paper introduces "Emu", a multimodal foundation model adept at generating both images and text in a multimodal context. Unlike traditional models that focus solely on text, Emu can accept various data inputs, including image, text, and video, through a unified autoregressive training process. Visual data is converted into embeddings, which, when combined with text tokens, form an integrated input sequence. The model's training objective is to either classify the next text token or regress the next visual embedding in this sequence. Emu's strength lies in its ability to utilize a wide range of pretraining data sources, such as videos combined with text or web pages with intermingled images and text. This versatility makes Emu suitable for tasks like image captioning, visual question answering, and text-to-image generation, where it outperforms other leading multimodal models. Emu's advanced features include in-context generation of text and images, image blending, video comprehension, and knowledge grounding. The model's effectiveness is further showcased as a multimodal assistant that can interact with users using both text and visuals.

**Strengths:**

The model shows ability to do versatile generation and strong in-context learning capability.

**Weaknesses:**

Certain model details is not clear:
1. How does causal transformer convert an image as multiple visual tokens. In section 2, is {z_1, z_2, ... z_N} the same as g(I).
2. Is the N visual embeddings for the image decoder the same as the visual ebmedding after the causal transformer.

Potential data issue impact the "zero-shot" restult:
EMU is trained with Laion-COCO, which has image caption in the style of COCO. How does that impact results in table 1? Is the zero-shot results as good if removing Laion-COCO?

Miss baselines:
Other multimodal baselines for FID such as https://arxiv.org/abs/2309.02591 should be added to table 2.

Suboptimal image generation capbility:
As shown in table2, there is performance loss due to regression to visual embeddings, instead of directly optimizing generating the best images.

**Questions:**

what is the FID before and after the visual decoding training?

---

> ### Author Response · Authors · 2023-11-20
> **Response to Reviewer qjEg (1/2)**
>
> We thank the reviewer for the positive comments. Below we address each question in detail.
>
> **Q1: Model Details Elaboration**
>
> **Mechanism of Causal Transformer.**
>
> {z_1, z_2, ... z_N} is not the same as g(I). g(I) is the encoding of images, outputted from EVA-CLIP, which will be fed into the Causal Transformer to be transformed into causal tokens {z_1, z_2, ... z_N}. They are the input and output of Causal Transformer, respectively.
>
> Each block of Causal Transformer consists of three layers: causal (masked) self-attention, cross-attention, and feed-forward layer. The process of aggregating information from 2D spatial visual signals into 1D causal sequence layer by layer is elaborated below:
>
> - Causal self-attention layer: we feed N learnable tokens (vectors) into this layer as Q, K and V. The output tokens encode causal dependencies among each other.
> - Cross-attention layer: the outputs from the previous causal self-attention layer serve as Q, while image features (g(I)) from EVA-CLIP serve as K and V. Each token can see all image features and extract information from them.
> - The feed-forward layer: a standard feed-forward layer.
>
> Through this process, the input vectors aggregate information from the image features while encoding causal dependency among each other, becomes {z_1, z_2, ... z_N}.
>
> **Is the N visual embeddings for the image decoder the same as those after the causal transformer?** No. The input of the image decoder is the embeddings generated by the Multimodal Modeling module.
>
> **Q2: Zero-shot concern about COCO evaluation**
>
> We thank the reviewer for the comment. We first report the zero-shot results on NoCaps and Flickr30K captioning benchmarks, which are absolutely out-of-domain and render no concerns about zero-shot. The results show that Emu achieves superior performance over the baselines.
>
> |             | NoCaps    | Flickr30K |
> | ----------- | --------- | --------- |
> | MetaLM      | 58.7      | 43.3      |
> | KOSMOS-1    | -         | 67.1      |
> | Flamingo-9B | -         | 61.5      |
> | Emu         | 96.5      | 72.0      |
> | Emu-I       | **108.8** | **77.4**  |
>
> We report the results of removing LAION-COCO below.
> |                                           | COCO Cap  |
> | ----------------------------------------- | --------- |
> | w LAION COCO                              | 112.9     |
> | w/o LAION COCO                            | 89.1      |
> | w 20M GPT-reformulated data from LAION-2B | **117.0** |
>
> The w/o LAION COCO experiment contains image-text pairs from only LAION-2B. LAION-2B is a large-scale dataset rich in world knowledge, but it contains poor captions, as exemplified by instances such as 'Sentry Gaming Headset, HPX-GX250B' and 'Angry Birds 2 Level 107 Shangham 3-Star Walkthrough'. Models pretrained exclusively on LAION-2B tend to generate text outputs of a similar nature, leading to a drop in CIDEr on the COCO caption benchmark, which specifically emphasizes the well-formed text. To encourage the learning of well-formed text, we included the synthetic dataset LAION-COCO during pretraining.
>
> LAION-COCO serves as a valuable supplement to LAION-2B by providing examples of well-formed captions, thereby aiding models in generating more coherent and linguistically appropriate text. It's worth noting that LAION-COCO can be replaced by other datasets featuring well-formed text. For instance, by randomly sampling 20 million samples from LAION-2B and rewriting the captions using ChatGPT, the model exhibits a notable improvement in zero-shot performance on the COCO caption benchmark, even without LAION-COCO. This demonstrates the adaptability of the model to different well-formed text datasets for enhanced performance.
>
> **Q3: Miss baselines.**
>
> We thank the reviewer for pointing it out. We have incorporated a discussion and result comparison in our updated paper (Section D). It is noteworthy that our project was completed prior to our awareness of the contemporaneous work, CM3Leon [1], which was released to arXiv less than one month before the ICLR submission deadline.
>
> In addition to the paper revision, we provide a comparison here. We can find that Emu outperform CM3Leon by a significant margin on most of tasks.
>
> |                   | COCO FID  | COCO Caption | VQAv2    | OKVQA    | VizWiz   | VisDial  |
> | ----------------- | --------- | ------------ | -------- | -------- | -------- | -------- |
> | CM3Leon (Sept. 5) | **10.87** | 61.6         | 47.6     | 23.8     | **37.6** | 22.6     |
> | Emu               | 11.66     | **112.4**    | **52.9** | **42.8** | 34.4     | **47.8** |
>
> [1] Scaling Autoregressive Multi-Modal Models: Pretraining and Instruction Tuning

---

> ### Author Response · Authors · 2023-11-20
> **Response to Reviewer qjEg (2/2)**
>
> **Q4: Suboptimal image generation capability and performance loss in Table 2.**
>
> We thank the reviewer for the question. We would like to clarify that this is probably because the condition space (image embeddings) of our visual decoder deviates a lot from the condition space (text embeddings) of the diffusion model used as initialization. Besides, our model is finetuned for a relatively short 15k steps, as discussed in Section 5.1.
>
> **Q5:  the FID before and after visual decoder training.**
>
> The FID before visual decoding training is not meaningful because the condition space (image embeddings) of our visual decoder deviates a lot from the condition space (text embeddings) of the diffusion model used as initialization. Moreover, we can not calculate the FID directly without visual decoder training because of dimension mismatch between our image embeddings and text embeddings used in the original diffusion model. We report the intermediate FID in the training progress before the complete training here:
>
> |           | COCO FID |
> | --------- | -------- |
> | 2k steps  | 17.05    |
> | 5k steps  | 15.74    |
> | 15k steps | 11.66    |

---

> ### Author Response · Authors · 2023-11-22
> **Anticipating Your Participation as Reviewer-Author Discussion Deadline Approaching**
>
> We express our sincere gratitude to the reviewer for dedicating time to review our paper. **We have provided comprehensive responses to all the questions including 1) the mechanism of our model, 2) zero-shot evaluation, 3) missed baselines, and 4) the performance of visual decoder**. As the discussion deadline looms within a day, we would like to inquire if our responses have adequately addressed your questions. We anticipate your participation in the Reviewer-Author discussion phase, as your insights are invaluable for refining and enhancing the quality of our paper. Should you have any additional queries or require further clarification, please do not hesitate to inform us. We are more than willing to address any concerns and ensure a comprehensive resolution. Thank you for your time and consideration.

---

> > ### Comment · Reviewer_qjEg · 2023-11-23
> > **Response to authors**
> >
> > Thanks for your feedback and additional results. I would like to keep my original rating.

---

### Official Review · Reviewer_yynn · 2023-11-03

**Soundness:** 2 fair
**Presentation:** 4 excellent
**Contribution:** 3 good
**Rating:** 6
**Confidence:** 4

**Summary:**

The paper presents a novel large multimodal model, Emu that takes as input the interleaved visual and textual data and generates images and text. The key contributions of the paper are three-folds: leveraging web-scale video data as a new source of interleaved data, proposing a novel objective of predicting-the-next-element in an unified autoregressive manner, and the model architecture enabling learning under such objective. The pretrained, or instruct-tuned models show remarkable performance on diverse multimodal tasks, which are shown qualitatively and quantitatively in the paper.

**Strengths:**

+ I agree with that the importance of video as a data source for learning large multimodal models has been overlooked so far. Leveraging videos as interleaved data will definitely provide much diverse supervision signals and facilitate scaling up training data
+ The writing and presentation of the paper is really good. The paper is overall well-written and easy to read. Especially, the authors describe all the details of the model architecture and training/inference procedures.
+ The experimental results validate the effectiveness of the proposed method.

**Weaknesses:**

Emu applies the regression loss to latent embeddings computed by the Causal Transformer, whose parameters are randomly initialized and also learned during pretraining. I was surprised that the training went well with the proposed objective, because I think that without additional constraints, the model may easily fall into a degenerate case, like the Causal Transformer always outputting constant vectors. Please elaborate on the mechanism of the proposed l2 regression loss.

**Questions:**

I am not sure about the composition of each mini-batch: does each mini-batch comprise heterogeneous samples with a batch size of 128+128+64+16+16=352, or homogeneous samples with a batch size of at most 128? And the authors said that they pretranied Emu only for 10k steps while the total number of training samples is 82M. This means that the model has seen at most 4.3% of the training data during pretraining. Is this a typo? otherwise, please explain how it was possible to learn so well with such a small amount of data.

---

> ### Author Response · Authors · 2023-11-20
> **Response to Reviewer yynn**
>
> We thank the reviewer for the positive feedback. Below we address the comments in detail.
>
> **Q1: The mechanism of the proposed L2 regression loss**
>
> The process can be similar to the bootstrapping mechanism [1] in self-supervised learning which works well with appropriate data strategy and pretraining tasks. Taking contrastive learning as an example, at the beginning of training, the model is randomly initialized and outputs random vectors, but the learning objective is to push the positive pairs closer. Such bootstrapping (predicting models' own output) can yield great visual representation although starting from random vectors.
>
> During the generative pretraining, the l2 regression loss and text classification loss work simultaneously on the diversified and mixed data that represent different types of tasks and constraints on the visual embeddings. For example, image-text pairs require the model to complete the image captioning task, which prevents the model collapse. In addition, the strong image-text-aligned initialization for the visual encoder, i.e., EVA-CLIP, makes the visual embeddings easy to align with the language model.
>
> Should any aspects require additional clarification, please feel free to communicate them to us. We are willing to engage in further discussions. Thank you.
>
> [1] Bootstrap Your Own Latent: A New Approach to Self-Supervised Learning. NIPS 2020
>
> **Q2: Question about the composition of each batch and the quantity of training data.**
>
> We thank the reviewer for the question. We deliberately use a small portion of LAION-2B and LAION-COCO, while fully leveraging other datasets in our pretraining.
>
> We deliberately avoid using the complete LAION-2B and LAION-COCO datasets, because LAION-2B includes a significant amount of noise and poorly-formed text. In contrast, LAION-COCO presents well-formed text but lacks the depth of world knowledge, containing relatively simpler text. We recommend [1] as a reference that analyses the deficiency of these two large-scale datasets in detail. To strike a balance between harnessing valuable information and mitigating the impact of noise during pretraining, we opted to use only a limited percentage of data.
>
> Specifically, we incorporated 32 million samples from LAION-2B and 32 million from LAION-COCO into the pretraining process, aiming for this careful equilibrium. Meanwhile, we fully leveraged other datasets in our pretraining. Emu was pretrained on 16 million samples from WebVid-10M(encompassing 10 million videos) and 4 million samples from MMC4 (a curated set of 2 million documents, selectively filtered based on image-text similarity threshold of >=0.32 ). Additionally, we've included 4 million samples from YT-Storyboard-1B, drawn from a pool of 18 million videos.
>
> Each mini-batch is composed of homogeneous samples from the same dataset, with a batch size of 128 for image-text pairs from LAION-2B and LAION-COCO, 64 for video-text pairs from WebVid-10M, 16 for interleaved image-text from MMC4, and interleaved video-text data from YT-Storyboard-1B.
>
> [1] CapsFusion: Rethinking Image-Text Data at Scale.

---

> ### Author Response · Authors · 2023-11-22
> **Anticipating Your Participation as Reviewer-Author Discussion Deadline Approaching**
>
> We express our sincere gratitude to the reviewer for dedicating time to review our paper. **We have provided comprehensive responses to all the questions including 1) the mechanism of the L2 regression loss, and 2) the composition of each batch and the quantity of training data.** As the discussion deadline looms within a day, we would like to inquire if our responses have adequately addressed your questions. We anticipate your participation in the Reviewer-Author discussion phase, as your insights are invaluable for refining and enhancing the quality of our paper. Should you have any additional queries or require further clarification, please do not hesitate to inform us. We are more than willing to address any concerns and ensure a comprehensive resolution. Thank you for your time and consideration.

---

> > ### Comment · Reviewer_yynn · 2023-11-22
> > **Thank you for the rebuttals.**
> >
> > I appreciate the efforts of the authors to provide rebuttals. It makes sense that training the model with both language modeling and l2 regression prevents the model from collapsing. However, I am not sure whether we can consider the training objective of Emu similar to that of non-contrastive Siamese networks such as BYOL and SimSiam [1]; these networks use explicit constraints, such as stop gradient (or momentum encoders) to prevent collapsing.
> >
> > [1] Xinlei Chen and Kaming He, Exploring Simple Siamese Representation Learning, CVPR 2021.

---

> ### Author Response · Authors · 2023-11-22
> **Thank you for the comments.**
>
> Thank you for your insightful comment! The joint learning of language modeling and l2 regression plays a crucial role in preventing the model from collapsing. Furthermore, we hope to provide a broader perspective by comparing with the mentioned SimSiam and BYOL. In learning objective, Emu, SimSam and BYOL similarly involve predicting the model's own output. However, they use different constraints to prevent the model from collapsing, e.g., joint supervision, stop gradient, momentum encoders.
>
> Thank you again for your time and consideration. Please do not hesitate to inform us if further clarification is required. We are more than willing to address any concerns.

---

> > ### Comment · Reviewer_yynn · 2023-11-23
> > **Response to authors**
> >
> > Thank you for the clarification. I will keep my rating.

---

### Author Response · Authors · 2023-11-20
**Global Response to Reviewers**

We extend our sincere gratitude to both the reviewers and the area chairs for your dedicated time invested in reviewing our paper. We have diligently addressed all of the reviewers' concerns in the corresponding responses, and uploaded a revision of our paper. In summary, there are two common concerns regarding ethics issues (Reviewer qbfH and paGX) and baseline missing (Reviewer qjEg, qbfH).

$1)$ Ethics concerns

We have included a full page in our updated paper to discuss the potential risks and mitigation strategies of Emu.

Emu is currently in a preliminary stage and has been developed solely for research purposes. Its usage in specific applications is not recommended until comprehensive risk analyses have been conducted. The ensuing discussion outlines potential risks and corresponding mitigation strategies of Emu, acknowledging the necessity for further research efforts to comprehensively assess associated risks.

The ethical considerations associated with Emu primarily stem from two key aspects:

1. Model initialization: the Multimodal Modeling module of Emu is initialized from LLaMA, the Visual Decoder module is initialized from Stable Diffusion, and the Vision Encoder is initialized from EVA-CLIP. Consequently, Emu inherits the potential risks of generating harmful and biased information, including offensive language, propagation of social biases and stereotypes, and the generation of inappropriate content such as pornography and child abuse.

2. Pretraining data. The pretraining data of Emu are publicly available and they are sourced from the Internet, where bias and harmful information are prevalent. Besides, the datasets sourced from the Internet (such as Common Crawl) may include links to images with personal information, potentially compromising privacy and containing sensitive content like faces, medical images, or other personal data.

It is crucial to reiterate that Emu is designed exclusively for preliminary academic research and should not be deployed in specific applications without rigorous risk analyses and mitigation strategy exploration. Deployment in production environments warrants a more thorough investigation into model behavior and potential biases.

Given the extensive size of pre-training datasets and the associated training costs, curating datasets and developing models for widespread use exceeds the scope of a single research paper. However, we are open to discussing mitigation strategies to help address ethical concerns.

Short-term approaches include: 1) relying on prompting to mitigate biases and harmful outputs, 2) implementing rule-based filtering to identify and block harmful information, 3) employing a discriminator model capable of classifying harmful information for enhanced blocking, 4) Emu itself can be finetuned to become a multimodal discriminator.

In the long term, strategies involve: 1) social or public policy interventions, such as regulatory frameworks and guidelines; 2) thoughtful product design, especially regarding user interface decisions; 3) advancements in AI Ethics of powerful large models, including the development of better benchmarks and improved mitigation strategies.

Additionally, to address privacy concerns, methods exist for obfuscating or generating personal human attributes like faces, ensuring anonymity without compromising the quality of learned representations. While this avenue is worth exploring, it is currently beyond the scope of this paper.

In conclusion, Emu is presently a model intended for preliminary research purposes only, and deployment should be deferred until the aforementioned issues are thoroughly considered and addressed.

$2)$ Baseline Missing

We have added the discussion and results of baselines to our revision (Section D) following reviewers' advice. We would like to clarify that the mentioned papers were released to arXiv in less than one month before the ICLR submission deadline, e.g., Sept 5 for Cm3Leon [1], Sept 27 for AnyMAL [2], and Sept 28 for ICLR. In fact, we have already finished our project before seeing the mentioned concurrent work. But we still thank the reviewer for pointing it out, which will give the readers a bigger picture.

Other concerns about model mechanism (Reviewer yynn, qjEg), training details (Reviewer yynn), evaluation (Reviewer qjEg, paGX), scalability (Reviewer qbfH) and additional limitations (Reviewer qbfH, qbfH) are all addressed accordingly in the respective rebuttals. We express our sincere appreciation again to the reviewers and the area chairs for your efforts in reviewing our work.

[1] Scaling Autoregressive Multi-Modal Models: Pretraining and Instruction Tuning

[2] AnyMAL: An Efficient and Scalable Any-Modality Augmented Language Model

---

### Meta-Review · Area_Chair_JmDU · 2023-12-10

**Metareview:**

This paper presents a new multimodal LLM that takes interleave image and text as input and generate image and text as output. The interleaved format enables a unified training object of predicting next token in an autoregressive way, thus also enables the image-text understanding and generation in one single model. This work also proposes to leverage diverse pre-training data sources, including videos and multimodal webpages.

The pros of the paper from reviewers:
1. A novel multimodal LLM unifies image and text input in an interleaved format.
2. Strong performance on a wide variety of tasks from image-text understanding and generation.
3. The writing and presentation is very clear, and the paper is easy to follow.

There are concerns about scalability, ethics issues, and comparison to concurrent work. The authors actively participated in the rebuttal phase and made revisions with more discussion and new results of them.

Overall, this paper presents clearly a new multimodal LLM, and demonstrates its strong performance on a varieties of image-text understanding and generation tasks.

**Justification For Why Not Higher Score:**

Although the scores are all positive, non of the reviewers to be too excited about it.

**Justification For Why Not Lower Score:**

It looks like a clear acceptance with all reviewers leaning towards positive.

---

### Decision · Program_Chairs · 2024-01-16

Accept (poster)